# Expressive Sign Equivariant Networks for Spectral Geometric Learning

**Derek Lim**
MIT CSAIL
dereklim@mit.edu

**Joshua Robinson**[*]
Stanford University

**Stefanie Jegelka**
TU Munich, MIT CSAIL

**Haggai Maron**
Technion, NVIDIA

## Abstract

Recent work has shown the utility of developing machine learning models that respect the structure and symmetries of eigenvectors. These works promote sign invariance, since for any eigenvector $v$ the negation $-v$ is also an eigenvector. However, we show that sign invariance is theoretically limited for tasks such as building orthogonally equivariant models and learning node positional encodings for link prediction in graphs. In this work, we demonstrate the benefits of sign *equivariance* for these tasks. To obtain these benefits, we develop novel sign equivariant neural network architectures. Our models are based on a new analytic characterization of sign equivariant polynomials and thus inherit provable expressiveness properties. Controlled synthetic experiments show that our networks can achieve the theoretically predicted benefits of sign equivariant models.

## 1   Introduction

The need to process eigenvectors is ubiquitous in machine learning and the computational sciences. For instance, there is often a need to process eigenvectors of operators associated with manifolds or graphs [Belkin and Niyogi, 2003, Rustamov et al., 2007], principal components (PCA) of arbitrary datasets [Pearson, 1901], and eigenvectors arising from implicit or explicit matrix factorization methods [Levy and Goldberg, 2014, Qiu et al., 2018]. However, eigenvectors are not merely unstructured data—they have rich structure in the form of symmetries [Ovsjanikov et al., 2008].

Specifically, eigenvectors have sign and basis symmetries. An eigenvector $v$ is sign symmetric in the sense that the sign-flipped vector $-v$ is also an eigenvector of the same eigenvalue. Basis symmetries occur when there is a repeated eigenvalue, as then there are infinitely many choices of eigenvector basis for the same eigenspace. Prior work has developed neural networks that are invariant to these symmetries, improving empirical performance in several settings [Lim et al., 2023].

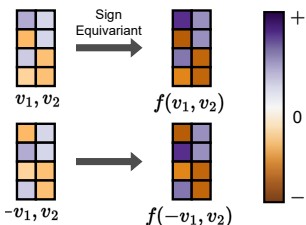

Figure 1: Illustration of a sign equivariant function $f$. When column 1 of the input is negated, column 1 of the output is also negated.

The goal of this paper is to demonstrate why sign *equivariance* can be useful and to characterize fundamental expressive sign equivariant architectures. Our first contribution is to show that sign equivariant models are a natural choice for several applications, whereas sign *invariant* architectures are provably insufficient for these applications. First, we show that sign and basis invariant networks are theoretically limited in expressive power for learning edge representations (and more generally multi-node representations) in graphs because they learn structural node embeddings that are known to be limited for link prediction and multi-node tasks [Srinivasan and

---

[*]Work completed whilst at MIT.

37th Conference on Neural Information Processing Systems (NeurIPS 2023).

Ribeiro, 2019, Zhang et al., 2021]. In contrast, we show that sign equivariant models can bypass this limitation by maintaining positional information in node embeddings. Furthermore, we show that sign equivariance combined with PCA can be used to parameterize expressive orthogonally equivariant point cloud models, thus giving an efficient alternative to PCA-based frame averaging [Puny et al., 2022, Atzmon et al., 2022]. In contrast, sign invariant models can only parameterize orthogonally *invariant* models in this framework, which excludes many important application areas.

The second contribution of this work is to develop the first sign equivariant neural network architectures, with provable expressiveness guarantees. We first present a difficulty in developing sign equivariant models: the "Geometric Deep Learning Blueprint" [Bronstein et al., 2021] suggests developing an equivariant neural network by interleaving equivariant *linear* maps and equivariant elementwise nonlinearities [Cohen and Welling, 2016, Ravanbakhsh et al., 2017, Finzi et al., 2021], but we show that our attempts to apply this approach are insufficient for expressive sign equivariant models. Namely, we show that sign equivariant linear maps between various input and output representations are very limited in their expressive power.

Hence, to develop our models, we derive a complete characterization of the sign equivariant polynomial functions. The form of these equivariant polynomials directly inspires our equivariant neural network architectures. Further, our architectures inherit the theoretical expressive power guarantees of the equivariant polynomials. Our characterization is also broadly useful for analysis and development of sign-symmetry-respecting architectures—for instance, we provide a new proof of the universality of SignNet [Lim et al., 2023] by showing that it can approximate all sign invariant polynomials.

To validate our theoretical results, we conduct various numerical experiments on synthetic datasets. Experiments in link prediction, n-body problems, and node clustering in graphs support our theory and demonstrate the utility of sign equivariant models.

## 1.1 Background

Let $f : \mathbb{R}^{n \times k} \to \mathbb{R}^{n \times k}$ be a function that takes eigenvectors $v_1, \ldots, v_k \in \mathbb{R}^n$ of an underlying matrix as input, and outputs representations $f(v_1, \ldots, v_k)$. We often concatenate the eigenvectors into a matrix $V = [v_1, \ldots, v_k] \in \mathbb{R}^{n \times k}$, and write $f(V)$ as the application of $f$. For simplicity, in this work we assume the eigenvectors come from a symmetric matrix, so they are taken to be orthonormal.

**Sign and basis symmetries.** Eigenvectors have symmetries, because there are many possible choices of eigenvectors of a matrix. For instance, if $v$ is a unit-norm eigenvector of a matrix, then so is the sign-flipped $-v$. If the eigenvalue of $v$ is simple, then $-v$ is the only other choice of unit-norm eigenvector of this eigenvalue.

If $v_1, \ldots v_m$ are an orthonormal basis of eigenvectors for the same eigenspace (meaning they all have the same eigenvalue), then there are infinitely many other choices of orthonormal basis for this eigenspace; these other choices of basis can be written as $VQ$, where $V = [v_1 \ldots v_m] \in \mathbb{R}^{n \times m}$ and $Q \in O(m)$ is an arbitrary orthogonal matrix.

We refer to these symmetries collectively as sign and basis symmetries, or more simply as eigenvector symmetries. Note that sign symmetries are a special case of basis symmetries, as $-1$ and $1$ are the only orthogonal $1 \times 1$ matrices. Previous work has developed neural networks that are invariant to these symmetries—that is, networks that have the same output for any choice of sign or basis of the eigenvector inputs [Lim et al., 2023].

**Sign equivariance** means that if we flip the sign of an eigenvector, then the corresponding column of the output of a function $f$ has its sign flipped. In other words, for all choices of signs $s_1, \ldots, s_k \in \{-1, 1\}^k$,

$$f(s_1 v_1, \ldots, s_k v_k)_{:,j} = s_j f(v_1, \ldots, v_k)_{:,j}, \tag{1}$$

where $A_{:,j}$ is the $j$-th column of an $n \times k$ matrix $A$. See Figure 1 for an illustration. In matrix form, letting $\mathrm{diag}(\{-1, 1\}^k)$ represent all $k \times k$ diagonal matrices with $-1$ or $1$ on the diagonal, $f$ is sign equivariant if

$$f(VS) = f(V)S \qquad \text{for all } S \in \mathrm{diag}(\{-1, 1\}^k). \tag{2}$$

As $O(1) = \{-1, 1\}$, we can write sign equivariance as equivariance with respect to a direct product of orthogonal groups $O(1) \times \ldots \times O(1)$. This is different from the equivariance to a single orthogonal group $O(d)$ considered in works on Euclidean-group equivariant networks [Thomas et al., 2018].

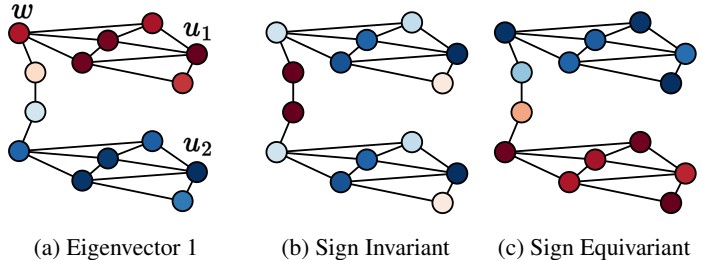

| (a) Eigenvector 1 | (b) Sign Invariant | (c) Sign Equivariant |

Figure 2: (a) First nontrivial normalized Laplacian eigenvector of a graph, which is positional. Nodes $u_1$ and $u_2$ are far apart in the graph, but automorphic. (b) Sign invariant node features, which are structural. Nodes $u_1$ and $u_2$ have the same feature. (c) Sign equivariant node features, which are positional. Nodes $u_1$ and $u_2$ have opposite signs. A link prediction model with sign invariant node features assigns $u_1$ and $u_2$ the same probability of connecting to $w$, while sign equivariant node features could give higher probability to $u_1$.

**Permutation equivariance** is often also a desirable property of our functions $f$. We say that $f$ is *permutation equivariant* if $f(PV) = Pf(V)$ for all $n \times n$ permutation matrices $P$. For instance, eigenvectors of matrices associated with simple graphs of size $n$ have such permutation symmetries, as the ordering of nodes is arbitrary.

## 2 Applications of Sign Equivariance

In this section, we present several applications for which modeling networks with sign equivariant architectures is beneficial. We identify that sign invariant networks are *provably* insufficient for these tasks, motivating the development of sign equivariant networks to address these limitations.

### 2.1 Multi-Node Representations and Link Prediction

In several settings, we desire a machine learning model that computes representations for tuples of several nodes in a graph. For instance, link prediction tasks generate probabilities for pairs of nodes, and both hyperedge prediction and subgraph prediction tasks learn representations for collections of nodes in a graph [Alsentzer et al., 2020, Wang et al., 2023]. For ease of exposition, the rest of this section discusses link prediction, though the discussion applies to general multi-node tasks as well.

In link prediction, we typically want to learn *structural node-pair representations*, meaning adjacency-permutation equivariant functions that give a representation for each pair of nodes; more precisely, a structural node-pair representation is a map $f : \mathbb{R}^{n \times n} \to \mathbb{R}^{n \times n}$ such that $f(PAP^\top) = Pf(A)P^\top$, where $f(A)_{i,j}$ is the representation of the pair of nodes $(i, j)$ in the graph with adjacency matrix $A$ [Srinivasan and Ribeiro, 2019] (see Appendix A.5 for more discussion). One method to do this is to use a graph model such as a standard GNN to learn node representations $z_i$, and then obtain a node-pair representation for $(i, j)$ as some function $f_{\text{decode}}(z_i, z_j)$ of $z_i$ and $z_j$. However, this approach is limited because standard GNNs learn *structural node encodings*—that is, adjacency-permutation equivariant node features $f_{\text{node}} : \mathbb{R}^{n \times n} \to \mathbb{R}^n$ such that $f_{\text{node}}(PAP^\top) = Pf_{\text{node}}(A)$ [Srinivasan and Ribeiro, 2019, Zhang et al., 2021]. [2] Structural node encodings give automorphic nodes the same representation, which can be problematic since automorphic nodes can be far apart in the graph. For instance, in Figure 2, $u_1$ and $u_2$ are automorphic, so a link predictor based on structural node encodings would give both $u_1$ and $u_2$ the same probability of connecting to $w$, but one would expect $u_1$ to have a higher probability of connecting to $w$. Most state-of-the-art link prediction methods on the Open Graph Benchmark leaderboards [Hu et al., 2020] were deliberately developed to avoid the issues of structural node encodings.

One way to surpass the limitations of structural node encodings is to use *positional node embeddings*, which can assign different values to automorphic nodes. Intuitively, positional encodings capture information such as distances between nodes and global position of nodes in the graph (see [Srinivasan and Ribeiro, 2019] for a formal definition). Laplacian eigenvectors are an important example of node

---

[2]This adjacency permutation equivariance is different from permutation equivariance of general vectors as defined in Section 1.1.

positional embeddings that capture much useful information of graphs [Chung, 1997, Von Luxburg, 2007]. In Figure 2 (a), the first nontrivial Laplacian eigenvector captures cluster structure in the graph, and as such assigns $u_1$ and $u_2$ very different values.

**Pitfalls of sign and basis invariance.** When processing eigenvectors of matrices associated with graphs, invariance to the symmetries of the eigenvectors has been found useful [Dwivedi et al., 2022a, Lim et al., 2023], especially for graph classification tasks. However, we show that exact invariance to these symmetries *removes positional information*, and thus the outputs of sign invariant or basis invariant networks are in fact *structural node encodings* (see Appendix A.5).[3] Hence, eigenvector-symmetry-invariant networks cannot learn node representations that distinguish automorphic nodes, and thus face the aforementioned difficulties when used for link prediction or multi-node tasks:

**Proposition 1.** *Let* $f : \mathbb{R}^{n \times k} \to \mathbb{R}^{n \times d_{\text{out}}}$ *be a permutation equivariant function, and let* $V = [v_1, \ldots, v_k] \in \mathbb{R}^{n \times k}$ *be* $k$ *orthonormal eigenvectors of an adjacency matrix* $A$. *Let nodes* $i$ *and* $j$ *be automorphic, and let* $z_i$ *and* $z_j \in \mathbb{R}^{d_{\text{out}}}$ *be their embeddings, i.e, the* $i$th *and* $j$th *row of* $Z = f(V)$.

- *If* $f$ *is sign invariant and the eigenvalues associated with the* $v_l$ *are simple and distinct, then* $z_i = z_j$.

- *If* $f$ *is basis invariant and* $v_1, \ldots, v_k$ *are a basis for some number of eigenspaces of* $A$ *then* $z_i = z_j$.

**A novel link prediction approach via sign equivariance.** The problem $z_i = z_j$ arises from the sign/basis invariances, which remove crucial positional information. We instead propose using sign *equivariant* networks (as in Section 3) to learn node representations $z_i = f(V)_{i,:} \in \mathbb{R}^k$. These representations $z_i$ maintain positional information for each node thanks to preserving sign information (see Figure 2 (c)). Then we use a sign invariant decoder $f_{\text{decode}}(z_i, z_j) = f_{\text{decode}}(Sz_i, Sz_j)$ for $S \in \text{diag}(\{-1, 1\}^k)$ to obtain node-pair representations. For instance, the commonly used $f_{\text{decode}} = \text{MLP}(z_i \odot z_j)$, where $\odot$ is the elementwise product, is sign invariant. When the eigenvalues are distinct, this approach has the desired invariances (yielding structural node-pair representations) and also maintains positional information in the node embeddings; see Appendix A.5 for a proof of the invariances, and Appendix A.5.1 for an example of where sign equivariant models can be used to compute strictly more expressive node-pair representations than sign invariant models. More details and the proof of Proposition 1 are in Appendix A.4.

Our sign equivariance based approach differs substantially from existing methods for learning structural pair representations without being bottlenecked by structural node representations. Many of these methods are based on labeling tricks [Zhang et al., 2021, Wang et al., 2023], whereby the representation for a node-pair is obtained by labeling the two nodes in the pair and then processing an enclosing subgraph. Without special modifications [Zhu et al., 2021, Chamberlain et al., 2023], this requires a separate expensive subgraph extraction and forward pass for each node-pair. In contrast, our method only requires one forward pass on the original graph to compute all positional node embeddings, after which pair representations can be obtained with a cheap, parallelizable decoding.

## 2.2 Orthogonal Equivariance

For various applications in modelling physical systems, we desire equivariance to rigid transformations; thus, orthogonally equivariant models have been a fruitful research direction in recent years [Thomas et al., 2018, Weiler et al., 2018, Anderson et al., 2019, Deng et al., 2021]. We say that a function $f : \mathbb{R}^{n \times k} \to \mathbb{R}^{n \times k}$ is orthogonally equivariant if $f(XQ) = f(X)Q$ for any $Q \in O(k)$, where $O(k)$ is the set of orthogonal matrices in $\mathbb{R}^{k \times k}$. Orthogonal equivariance imposes infinitely many constraints on the function $f$. Several works have approached this problem by reducing to a finite set of constraints using so-called Principal Component Analysis (PCA) based frames [Puny et al., 2022, Atzmon et al., 2022, Xiao et al., 2020].

PCA-frame methods take an input $X \in \mathbb{R}^{n \times k}$, compute orthonormal eigenvectors $R_X \in O(k)$ of the covariance matrix $\text{cov}(X) = (X - \frac{1}{n}\mathbf{1}\mathbf{1}^\top X)^\top (X - \frac{1}{n}\mathbf{1}\mathbf{1}^\top X)$ (assumed to have distinct eigenvalues), then average outputs of a base model $h$ for each of the $2^k$ sign-flipped inputs $XR_XS$, where $S \in \text{diag}(\{-1, 1\}^k)$. We instead suggest using a sign equivariant network to parameterize an efficient $O(k)$ equivariant model, which allows us to bypass the need to average the exponentially

---

[3] When there are repeated eigenvalues, sign invariant embeddings maintain some positional information.

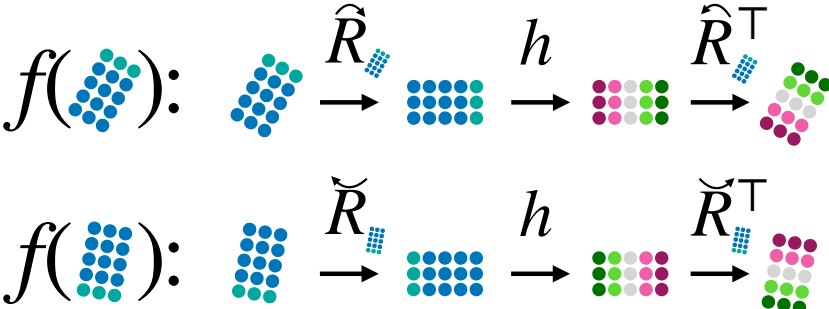

Figure 3: Using sign equivariant functions $h$ to parameterize orthogonally equivariant $f(X) = h(XR_X)R_X^\top$, where $R_X$ is a choice of principal components for the point cloud. We first transform $X$ via $R_X$ into an orientation that is unique up to sign flips, then process $XR_X$ using the sign equivariant model $h$, and finally reintegrate orientation information back into the output via $R_X^\top$.

Table 1: Sign invariant or equivariant polynomials and corresponding neural network architectures for different input and output spaces. $v \in \mathbb{R}^k$ or $V \in \mathbb{R}^{n \times k}$ are inputs to the polynomials or networks. Appendix C contains more details on the polynomials.

| Constraints | Polynomials | Neural Networks |
|---|---|---|
| $\mathbb{R}^k \to \mathbb{R}$ inv. | $\sum_{d_1,\ldots,d_k=0}^{D} \mathbf{W}_{d_1,\ldots,d_k} v_1^{2d_1} \cdots v_k^{2d_k}$ | $\mathrm{MLP}(|v|)$ |
| $\mathbb{R}^{n \times k} \to \mathbb{R}$ inv. | $q([V_{i_1,j} \cdot V_{i_2,j}]_{i_1 \in [n], i_2 \in [n], j \in [k]})$ | $\mathrm{SignNet}(V) = \rho([\phi(v_i) + \phi(-v_i)]_{i=1,\ldots,k})$ |
| $\mathbb{R}^k \to \mathbb{R}^k$ equiv. | $v \odot p_{\mathrm{inv}}(v)$ | $v \odot \mathrm{MLP}(|v|)$ |
| $\mathbb{R}^{n \times k} \to \mathbb{R}^{n' \times k}$ equiv. | $W^{(2)}\left((W^{(1)}V) \odot p_{\mathrm{inv}}(V)\right)$ | $[W_1^{(l)}v_1, \ldots, W_k^{(l)}v_k] \odot \mathrm{SignNet}_l(V)$ |

many sign-flipped inputs. For a sign equivariant network $h$, we define our model $f$ to be

$$f(X) = h(XR_X)R_X^\top. \tag{3}$$

See Figure 3 for an illustration. Intuitively, this first transforms $X$ by $R_X$ into a nearly canonical orientation that is unique up to sign flips; this can be seen as writing the points in the principal components basis, or aligning the principal components of $X$ with the coordinate axes. Then we process $XR_X$ using the model $h$ that respects the sign symmetries, and finally, we incorporate orientation information back into the output by post-multiplying by $R_X^\top$. Our approach only requires one forward pass through $h$, whereas frame averaging requires $2^k$ forward passes through a base model. The following proposition shows that $f$ is $O(k)$ equivariant, and inherits universality properties of $h$.[4]

**Proposition 2.** *Consider a domain $\mathcal{X} \subseteq \mathbb{R}^{n \times k}$ such that each $X \in \mathcal{X}$ has distinct covariance eigenvalues, and let $R_X$ be a choice of orthonormal eigenvectors of $\mathrm{cov}(X)$ for each $X \in \mathcal{X}$. If $h : \mathcal{X} \subseteq \mathbb{R}^{n \times k} \to \mathbb{R}^{n \times k}$ is sign equivariant, and if $f(X) = h(XR_X)R_X^\top$, then $f$ is well defined and orthogonally equivariant.*

*Moreover, if $h$ is from a universal class of sign equivariant functions, then the $f$ of the above form universally approximate $O(k)$ equivariant functions on $\mathcal{X}$.*

We include a proof of this result in Appendix A.6. This result also follows from Theorems 3.1 and 3.3 of Kaba et al. [2023], who show that generally we can canonicalize up to a subgroup $K$ of a group $G$, and achieve $G$-equivariance via a $K$-equivariant base predictor. In our case, $G = O(k)$ and $K = \{-1, 1\}^k$

**Sign invariance only gives orthogonal invariance.** In a similar way, a sign *invariant* model can be used to obtain an orthogonally *invariant* model with PCA frames, but it cannot be used for orthogonal equivariance; instead, sign equivariance is needed.

# 3 Sign Equivariant Polynomials and Networks

In this section, we analytically characterize the sign equivariant polynomials, and use this characterization to develop sign equivariant architectures. As equivariant polynomials universally approximate continuous equivariant functions [Yarotsky, 2022], our architectures inherit universality guarantees. We summarize our results on polynomials and neural network architectures in Table 1. Our characterization of the invariant polynomials also allows us to give an alternative proof of the universality of the sign invariant neural network SignNet [Lim et al., 2023] (see Appendix C.4).

## 3.1 Sign Equivariant Linear Maps

First, we consider the important case of degree one polynomials, i.e. sign equivariant linear maps from $\mathbb{R}^{n \times k} \to \mathbb{R}^{n' \times k}$. These maps are very limited in expressive power, as they act independently on each eigenvector.

**Lemma 1.** *A linear map $W : \mathbb{R}^{n \times k} \to \mathbb{R}^{n' \times k}$ is sign equivariant if and only if it can be written as*

$$W(X) = [W_1 X_1 \ \ldots \ W_k X_k] \tag{4}$$

*for some linear maps $W_1, \ldots, W_k : \mathbb{R}^n \to \mathbb{R}^{n'}$, where $X_i \in \mathbb{R}^n$ is the $i$th column of $X \in \mathbb{R}^{n \times k}$.*

See Appendix B.1 for the proof. Notably, when $n = n' = 1$, the linear maps are diagonal matrices.

This means a model with elementwise nonlinearities and sign equivariant linear maps will not capture any *interactions* between eigenvectors. For instance, when used for parameterizing orthogonally equivariant models as in Section 2.2, such a model would process each principal component direction of the point cloud independently. Hence, the popular approach —outlined in the "Geometric Deep Learning Blueprint" [Bronstein et al., 2021]—of interleaving equivariant linear maps and equivariant nonlinearities [Cohen and Welling, 2016, Zaheer et al., 2017, Kondor and Trivedi, 2018, Maron et al., 2018, 2019, Finzi et al., 2021] is not as fruitful here.

However, one may choose instead different group representations for the input and output space, but our attempts to do this do not lead to efficient models. For instance, a common method to improve expressive power of models that use equivariant linear maps is to use tensor representations [Maron et al., 2018, 2019, Finzi et al., 2021]; in our case, this would correspond having equivariant hidden representations in $\mathbb{R}^{n \times k^m}$ for some tensor order $m$. This is also inefficient, as we explain in Appendix **??**; we show that such an approach would have to lift to tensors of at least order 3, and that there are many sign equivariant linear maps between tensors of order 3. There is a possibility that some other group representations may allow the Geometric Deep Learning Blueprint to work better for sign equivariant networks, but we could not find any such representations.

For these reasons, we will now analyze the entire space of sign equivariant polynomials.

## 3.2 Sign Equivariant Polynomials

Consider polynomials $p : \mathbb{R}^{n \times k} \to \mathbb{R}^{n' \times k}$ that are sign equivariant, meaning $p(VS) = p(V)S$ for $S \in \text{diag}(\{-1, 1\}^k)$. We can show that a polynomial $p$ is sign equivariant if and only if it can be written as the elementwise product of a simple (linear) sign equivariant polynomial and a general sign invariant polynomial, followed by another linear sign equivariant map.

**Theorem 1.** *A polynomial $p : \mathbb{R}^{n \times k} \to \mathbb{R}^{n' \times k}$ is sign equivariant if and only if it can be written*

$$p(V) = W^{(2)} \left( (W^{(1)}V) \odot p_{\text{inv}}(V) \right) \tag{5}$$

*for sign equivariant linear $W^{(2)}$ and $W^{(1)}$, and a sign invariant polynomial $p_{\text{inv}} : \mathbb{R}^{n \times k} \to \mathbb{R}^{n' \times k}$.*

This reduction of sign equivariant polynomials to sign invariant polynomials combined with simple operations is convenient, as it enables us to leverage recent universal models for sign invariant functions [Lim et al., 2023]. The proof of this statement is in Appendix C, which proceeds by showing that sign equivariance leads to linear constraints on the coefficients of a polynomial, which requires the polynomial to take the form stated in the Theorem.

---

[4]A class of model functions $\mathcal{F}_{\text{m}}$ from $\mathcal{X} \to \mathcal{Y}$ is universal with respect to a target class $\mathcal{F}_{\text{t}}$ if for all compact $\mathcal{D} \subseteq \mathcal{X}$, $f_{\text{t}} \in \mathcal{F}_{\text{t}}$, and $\epsilon > 0$, there is an $f_{\text{m}} \in \mathcal{F}_{\text{m}}$ such that $\|f_{\text{m}}(x) - f_{\text{t}}(x)\| < \epsilon$ for all $x \in \mathcal{D}$.

Table 2: Link prediction AUC and runtime per epoch for structural edge models.

| | Erdős-Rényi | | Barabási-Albert | |
|---|---|---|---|---|
| Model | Test AUC | Runtime (s) | Test AUC | Runtime (s) |
| GCN (constant input) | .497±.06 | .058±.00 | .705±.01 | .048±.00 |
| SignNet | .498±.00 | .120±.00 | .707±.00 | .095±.00 |
| $V_{i,:}^\top V_{j,:}$ | .570±.01 | .010±.01 | .597±.01 | .008±.00 |
| $\text{MLP}(V_{i,:} \odot V_{j,:})$ | .614±.02 | .050±.00 | .651±.03 | .040±.00 |
| Sign Equivariant | **.751**±.00 | .063±.00 | **.773**±.01 | .054±.00 |

## 3.3 Sign Equivariance without Permutation Symmetries

Using Theorem 1, we can now develop sign equivariant architectures. We parameterize sign equivariant functions $f : \mathbb{R}^{n \times k} \to \mathbb{R}^{n' \times k}$ as a composition of layers $f_l$, each of the form

$$f_l(V) = [W_1^{(l)} v_1, \dots, W_k^{(l)} v_k] \odot \text{SignNet}_l(V), \tag{6}$$

in which the $W_i^{(l)} : \mathbb{R}^n \to \mathbb{R}^{n'}$ are arbitrary linear maps, and $\text{SignNet}_l : \mathbb{R}^{n \times k} \to \mathbb{R}^{n' \times k}$ is sign invariant [Lim et al., 2023]. In the case of $n = n' = 1$, there is a simple universal form: we can write a sign equivariant function $f : \mathbb{R}^k \to \mathbb{R}^k$ as $f(v) = v \odot \text{MLP}(|v|)$, where $|v|$ is the elementwise absolute value. These two architectures are universal because they can approximate sign equivariant polynomials. Here, the sign invariant part captures interactions between eigenvectors that the equivariant linear maps cannot.

**Proposition 3.** *Functions of the form $v \mapsto v \odot \text{MLP}(|v|)$ universally approximate continuous sign equivariant functions $f : \mathbb{R}^k \to \mathbb{R}^k$.*

*Compositions $f_2 \circ f_1$ of functions $f_l$ as in equation 6 universally approximate continuous sign equivariant functions $f : \mathbb{R}^{n \times k} \to \mathbb{R}^{n' \times k}$.*

## 3.4 Sign Equivariance and Permutation Equivariance

For models on eigenvectors that stem from graphs or point clouds, in addition to sign equivariance, we may demand permutation equivariance, i.e., $f(PV) = Pf(V)$ for all permutation matrices $P \in \mathbb{R}^{n \times n}$. To add permutation equivariance to our neural network architecture from Section 3.3, we use it within the framework of DeepSets for Symmetric Elements (DSS) [Maron et al., 2020]. For a hidden dimension size of $d_f$, each layer $f_l : \mathbb{R}^{n \times k \times d_f} \to \mathbb{R}^{n \times k \times d_f}$ of our DSS-based sign equivariant network takes the following form on row $i$:

$$f_l(V)_{i,:} = f_l^{(1)}(V_{i,:}) + f_l^{(2)}\Big(\sum\nolimits_{j \neq i} V_{j,:}\Big), \tag{7}$$

where $f_l^{(1)}$ and $f_l^{(2)}$ are sign equivariant functions as in Section 3.3. Sometimes we take $d_f = 1$, in which case we can use the simpler $\mathbb{R}^k \to \mathbb{R}^k$ sign equivariant networks ($v \odot \text{MLP}(|v|)$) as $f_l^{(1)}$ and $f_l^{(2)}$. If we have graph information, then we can do message-passing by changing the sum over $j \neq i$ to a sum over a neighborhood of node $i$. DSS has universal approximation guarantees [Maron et al., 2020], but they only apply for groups that act as permutation matrices, whereas the sign group $\{-1, 1\}^k$ does not. Hence, the universal approximation properties of our proposed DSS-based architecture are still an open question.

# 4 Experiments

Our theoretical results in Section 2 predict benefits of sign equivariance in various tasks: link prediction in nearly symmetric graphs, orthogonally equivariant simulations in n-body problems, and node clustering with positional information. Next, we probe these suggested benefits empirically.

## 4.1 Link Prediction in Nearly Symmetric Graphs.

We begin with a synthetic link prediction task, which is carefully controlled to test the theoretically foreseen benefits of sign equivariance explained in Section 2.1. With the intuition of Figure 2 we first

either generate an Erdős-Rényi [Erdős et al., 1960] or Barabási-Albert [Barabási and Albert, 1999] random graph $H$ of 1000 nodes. Then we form a larger graph $G$ that contains two disjoint copies of $H$, along with 1000 uniformly-randomly added edges (both between and within copies of $H$). Without the random edges, each node in one copy of $H$ is automorphic to the corresponding node in the other copy, so we expect many nodes to be nearly automorphic with the randomly added edges.

In Table 2, we show the link prediction performance of several models that learn structural edge representations. The methods that use eigenvectors have a sign invariant final prediction for each edge. GCN [Kipf and Welling, 2017] where the node features are all ones and SignNet [Lim et al., 2023] both completely fail on the Erdős-Rényi task (these two models map automorphic nodes to the same embedding), while our sign equivariant model outperforms all methods. We also try two eigenvector baselines that maintain node positional information, but do not update eigenvector representations: taking the dot product $V_{i,:}^\top V_{j,:}$ to be the logit of a link existing, or learning a simple decoder $\mathrm{MLP}(V_{i,:} \odot V_{j,:})$. Both perform substantially worse than our sign equivariant model, which shows that updating eigenvector representations is important here. Further, the sign equivariant model takes comparable runtime to GCN, and is significantly faster than SignNet. This is because we use networks of the form $v \mapsto v \odot \mathrm{MLP}(|v|)$ in these experiments instead of the full SignNet-based model in equation 6. See Appendix E.2 for more details.

## 4.2 Orthogonal Equivariance in n-body Problems

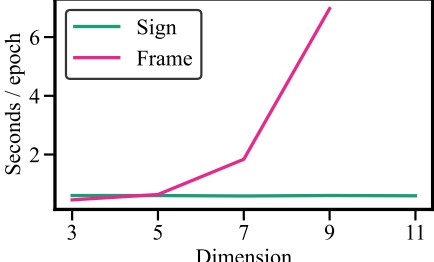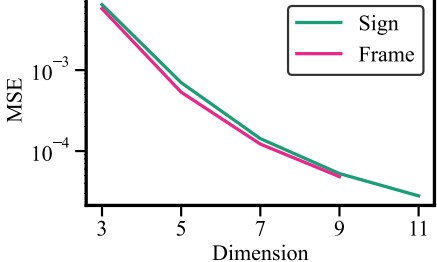

Figure 4: Sign equivariant model versus frame averaging model for n-body experiments in varying dimensions. Lower $y$-axis is better for both plots. (Left) The runtime of frame averaging increases exponentially in dimension while the sign equivariant runtime is approximately constant. Frame averaging runs out of memory on $d = 11$. (Right) The error of the sign equivariant model is very similar to that of frame averaging.

In this section, we empirically test the ability of our sign equivariant models to parameterize orthogonally equivariant functions on point clouds, as outlined in Section 2.2. For this purpose, we consider simulating n-body problems, following the setup in Fuchs et al. [2020] and building on the code from Puny et al. [2022]. To test the favorable scaling of our method in the dimension $d$ of the problem against the exponential $2^d$ scaling of frame averaging, we generalize this problem to general dimensions $d \geq 3$. We maintain the choice of $n = 5$ particles, and generate new point clouds using the same procedure as in Fuchs et al. [2020] (sampling random points and initial velocities in a general dimension $d$). We measure model performance via mean squared error (MSE). We use a DSS-based model that we describe in more detail in Appendix E.3.

Figure 4 illustrates the runtime and MSE. The sign equivariant model scales well with dimension—the time-per-epoch is nearly constant as we increase the dimension. In contrast, frame averaging suffers from the expected exponential slowdown with dimension, and runs out of memory on a 32GB V100 GPU for $d = 11$. Considering the MSE, the equivariant model's performance closely follows that of frame averaging, i.e., we only have a small loss in accuracy with much better scalability. For $d = 3$, the sign equivariant model has an MSE of .00646, compared to the .00575 of frame averaging [Puny et al., 2022]. Additional $d = 3$ comparisons to other baselines are included in Appendix E.3.

## 4.3 Node Clustering with Positional Information

As explained in Section 2.1, some applications on graph data call for positional node embeddings that can assign different representations to automorphic nodes. For instance, consider community detection or node clustering tasks on graphs, where a model makes a prediction for each node that

assigns it to a cluster. Structural encodings are insufficient for this task, as there may be automorphic or nearly-automorphic nodes that are far apart in the graph but look alike in a structural encoding. Hence, node structural encodings would guide the model to assign these nodes to the same cluster, even though they should belong to different clusters. As a concrete example, consider a graph of two clusters, and using Laplacian eigenvectors as positional encodings. The first nontrivial eigenvector will tend to assign a positive sign to one cluster and a negative sign to the other cluster. Thus, the sign information in the eigenvectors is crucial, so we expect sign equivariant models to perform well.

We test models on the CLUSTER dataset [Dwivedi et al., 2022a] for semi-supervised node clustering (viewed as node classification) in synthetic graphs. In these experiments, we build on the empirically well-performing GraphGPS model [Rampasek et al., 2022], and incorporate our sign equivariant models to update eigenvector representations within the version of GraphGPS that uses PEG [Wang et al., 2022] to process positional encodings. See Appendix E.4 for more experimental details.

As seen in Table 3, our sign equivariant models outperform all of the other GraphGPS-based eigenvector methods. Moreover, we achieve the second best performance across all methods, showing that sign equivariant models can indeed achieve the theoretically expected benefits in this setting.

Table 3: Results on the CLUSTER node classification task, for which positional information is needed. We compare different SOTA and Laplacian eigenvector-based methods.

| Model | Test Acc. (%) |
|---|---|
| GCN [Kipf and Welling, 2017] | $68.498_{\pm 0.976}$ |
| GIN [Xu et al., 2019] | $64.716_{\pm 1.553}$ |
| GAT [Veličković et al., 2018] | $70.587_{\pm 0.447}$ |
| GatedGCN [Bresson and Laurent, 2017] | $73.840_{\pm 0.326}$ |
| SAN [Kreuzer et al., 2021] | $76.691_{\pm 0.650}$ |
| K-Subgraph SAT [Chen et al., 2022] | $77.856_{\pm 0.104}$ |
| EGT [Hussain et al., 2022] | $\mathbf{79.232}_{\pm 0.348}$ |
| GPS [Rampasek et al., 2022] | $78.016_{\pm 0.180}$ |
| *Eigenvector Methods (GPS base model)* | |
| No PE | $77.423_{\pm 0.241}$ |
| LapPE [Dwivedi et al., 2022a] | $77.250_{\pm 0.280}$ |
| PEG [Wang et al., 2022] | $77.945_{\pm 0.310}$ |
| SignNet [Lim et al., 2023] | $77.442_{\pm 0.102}$ |
| Sign Equivariant (ours) | $\mathbf{78.201}_{\pm 0.118}$ |

## 5   Related Work

**Structural and Positional Representations.**   Especially for link prediction, the need for structural node-pair representations that are not obtained from structural node representations has been discussed in several works [Srinivasan and Ribeiro, 2019, Zhang et al., 2021, Cotta et al., 2023]. As such, various methods have been developed for learning structural node-pair representations that incorporate node positional information. SEAL and other labeling-trick based methods [Zhang and Chen, 2018, Zhang et al., 2021] use added node features depending on the node-pair that we want a representation of. This is empirically successful in many tasks, but typically requires a separate subgraph extraction and forward pass through a GNN for each node-pair under consideration. Distance encoding [Li et al., 2020] uses relative distances between nodes to capture positional information. PEG [Wang et al., 2022] similarly maintains positional information by using eigenvector distances between nodes in each layer of a GNN, but does not update eigenvector representations. Identity-aware GNNs [You et al., 2021] and Neural Bellman-Ford Networks [Zhu et al., 2021] learn pair representations by conditioning on a source node from the pair.

**Eigenvectors as Graph Positional Encodings.**   When using eigenvectors of graphs as node positional encodings for graph models like GNNs and Graph Transformers, many works have noted the need to address the sign ambiguity of the eigenvectors. This is often done by encouraging sign invariance through data augmentation—the signs of the eigenvectors are chosen randomly in each iteration of training [Dwivedi et al., 2022a,b, Kreuzer et al., 2021, Mialon et al., 2021, Kim et al., 2022, He et al., 2022, Müller et al., 2023]. In contrast, SignNet [Lim et al., 2023] enforces exact sign invariance, by processing eigenvectors with a sign invariant neural architecture; this approach has been taken by some recent works [Rampasek et al., 2022, Geisler et al., 2023, Murphy et al., 2023].

**Equivariant Neural Network Design.**   Equivariant neural network architectures have been proposed for various types of data and symmetry groups. A common paradigm is to interleave equivariant linear maps and equivariant pointwise nonlinearities [Wood and Shawe-Taylor, 1996, Cohen and Welling, 2016, 2017, Ravanbakhsh et al., 2017, Maron et al., 2018, Kondor and Trivedi, 2018, Finzi et al., 2021, Bronstein et al., 2021, Pearce-Crump, 2022]; this is often used when the group acts as some subset of the permutation matrices. However, the sign group does not act as permutation matrices, and as we explained above this approach is not expressive for sign equivariant models.

More similarly to our approach, many equivariant machine learning works heavily leverage invariant or equivariant polynomials (or other equivariant nonlinear functions). These works include polynomials as operations within a network [Thomas et al., 2018, Puny et al., 2023], add polynomials as features [Yarotsky, 2022, Villar et al., 2021], build networks that take a similar form to equivariant polynomials [Villar et al., 2021], and/or analyze neural network expressive power by determining which equivariant polynomials a given architecture can compute [Zaheer et al., 2017, Segol and Lipman, 2019, Maron et al., 2019, 2020, Chen et al., 2020, Dym and Maron, 2021, Puny et al., 2023].

## 6    Conclusion

In this work, we identify and study an important method of respecting the symmetries of eigenvector data—sign equivariant models. For multi-node representation tasks, link prediction, and orthogonally equivariant tasks, sign equivariance provides a natural inductive bias; in contrast, we show that sign invariant models are provably limited in these tasks. To develop sign equivariant neural networks, we analytically characterize the sign equivariant polynomials, and then define neural networks that parameterize functions of similar form. Our neural networks are thus expressive, and inherit universal approximation guarantees of the equivariant polynomials. In several experiments, we show that our neural networks can indeed achieve the theoretically predicted benefits of sign equivariant models.

**Limitations and Future Work.**    While we developed sign equivariant architectures in this work, we did not explore basis-change equivariant architectures, which would have the desired symmetries for inputs with repeated eigenvalues. As eigenvalue multiplicities are known to occur in many real-world graphs [Lim et al., 2023], future work in this area could be useful. Further, we give evidence that sign equivariance could help in some node-level and multi-node-level prediction tasks on graphs, but we do not have theoretical reason to believe that sign equivariance could help in graph-level representation tasks, which for instance are common in molecule processing. Our theoretical results are focused on expressive power, but we do not have results on other properties that are important for learning, such as optimization [Xu et al., 2021], stability [Wang et al., 2022, Huang et al., 2023], or generalization [Keriven and Vaiter, 2023]. Finally, while we can prove universality of our models in the non-permutation-equivariant setting, we do not know of the exact expressive power in the permutation equivariant setting. Lim et al. [2023] also faces this issue for sign invariant models; future work on analyzing and possibly improving the expressive power of these models — if they are not universal — is promising.

### Acknowledgments

We would like to thank Yaron Lipman for contributing significantly early on in this work. We would like to thank Maks Ovsjanikov for noting that basis invariant models give automorphic nodes the same representation, and also Johannes Lutzeyer and Michael Murphy for helpful comments. We would also like to thank the reviewers of the Physics4ML workshop at ICLR 2023 for helpful feedback and close reading. DL is supported by an NSF Graduate Fellowship. SJ acknowledges support from NSF AI Institute NSF CCF-2112665, NSF Award 2134108, and Office of Naval Research Grant N00014-20-1-2023 (MURI ML-SCOPE).

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

# A   Applications of Sign Equivariance

## A.1   Improving Invariant Eigenvector Networks

Neural networks that are invariant to eigenvector symmetries have been shown to empirically improve graph learning models and achieve theoretically high expressive power. SignNet [Lim et al., 2023], a sign invariant neural network, takes the form

$$f(v_1, \ldots, v_k) = \rho(\phi(v_1) + \phi(-v_1), \ldots, \phi(v_k) + \phi(-v_k)) \tag{8}$$

for neural networks $\rho$ and $\phi$. This directly enforces invariant representations, without any intermediate equivariant representations. However, many successful invariant models first have many equivariant layers before a final invariant operation as equivariant layers are more expressive: this includes convolutional neural networks [LeCun et al., 1989], message passing graph neural networks [Gilmer et al., 2017], invariant graph networks [Maron et al., 2018], and group convolutional neural networks [Cohen and Welling, 2016]. Thus, sign equivariant layers may lead to better sign invariant networks. Moreover, sign equivariant layers may improve on other aspects of SignNet, such as expressiveness of node features (Proposition 1) and efficiency (Appendix A.2)

## A.2   Efficiency Gains from Sign Equivariant Networks

Here, we show that our sign equivariant models can reduce the complexity of equivariant or invariant networks for two different types of applications. Throughout, we consider functions $f : \mathbb{R}^{n \times k} \to \mathbb{R}^{n \times k}$, and we consider our permutation equivariant and sign equivariant DSS-based architecture from Section 3.4.

The time cost (in floating point operations) per layer of our DSS-based model is $\mathcal{O}(n(kd + d^2))$, where $d$ is the maximum hidden dimension of the MLP and we assume constant depth MLPs. To see this, note that we can precompute $\sum_{j=1}^n V_{j,:}$, so that each $\sum_{j \neq i} V_{j,:}$ can be computed in constant time by subtracting $V_{i,:}$ from the total sum. Then for each of the $n$ rows, the MLPs require $\mathcal{O}(kd + d^2)$ to evaluate matrix multiplications. In this process, we only form tensors of size $\mathcal{O}(n(k + d))$, as the inputs and outputs are of size $\mathcal{O}(nk)$, and the hidden layers of the MLPs form tensors of size $\mathcal{O}(nd)$.

### A.2.1   Efficient Orthogonally Equivariant Networks

Consider the case of $O(k)$ equivariant models $f : \mathbb{R}^{n \times k} \to \mathbb{R}^{n \times k}$ such that $f(XQ) = f(X)Q$ for all orthogonal matrices $Q \in O(k)$. There are many orthogonally equivariant neural architectures that are specialized to the special case of $k = 3$, which is very useful for applications in the physical sciences [Thomas et al., 2018, Fuchs et al., 2020]. Here we consider models that directly work for general dimension $k$.

Frame averaging approaches [Puny et al., 2022, Atzmon et al., 2022] require $2^k$ forward passes of a base network $f_\theta$, one for each sign flip of the principal components. Letting their base network be a permutation equivariant DeepSets [Zaheer et al., 2017], this means that they require $\mathcal{O}(n(kd + d^2)2^k)$ time to evaluate their model, where $d$ is the hidden dimension of the base model. Note that this has an extra exponential $2^k$ factor compared to our $\mathcal{O}(n(kd + d^2))$ cost.

Another general approach with universality guarantees comes from Villar et al. [2021], who analyze invariant polynomials to develop equivariant architectures. However, their method for $O(k)$ invariance or equivariance requires forming $XX^\top$, an $n \times n$ matrix. Thus, the complexity is at least $\mathcal{O}(n^2)$, which is a problem in applications, since oftentimes $n$ is much larger than $k$. Variants of their method do not need to compute all $\mathcal{O}(n^2)$ inner products, but it is unclear how to maintain permutation equivariance when doing this.

### A.2.2   Efficient Sign Invariant Networks

Consider again the form of SignNet [Lim et al., 2023], $f(V) = \rho([\phi(v_i) + \phi(-v_i)]_{i=1,\ldots,k})$. In the permutation equivariant version, e.g. when $\phi$ is a DeepSets [Zaheer et al., 2017] or a message passing neural network [Gilmer et al., 2017], $\phi$ maps from $\mathbb{R}^n \to \mathbb{R}^{n \times d}$, where $d$ is the hidden dimension. Thus, computing $\phi(v_i) + \phi(-v_i)$ for all $k$ vectors $v_i$ require an $\mathcal{O}(nkd)$ sized tensor to be formed (even if the output space of $\phi$ is $\mathbb{R}^n$, a vectorized implementation computes all $\phi(v_i) + \phi(-v_i)$ in two batched inference calls to $\phi$, which would require $\mathcal{O}(nkd)$ sized intermediate tensors). This is

a multiplicative factor larger than the sign equivariant requirement of $\mathcal{O}(n(k+d))$ sized tensors. Moreover, it would take $\mathcal{O}(nkd^2)$ time to compute $\phi(v_i)+\phi(-v_i)$ for each $i$, which is a multiplicative factor larger than the $\mathcal{O}(n(kd+d^2))$ time for the sign equivariant architecture.

## A.3  Potential Societal Impacts

We do not foresee direct societal impacts from our work. This project is primarily theoretical and aims to improve models for two general application areas: multi-node representation learning and orthogonal equivariant models. Potential societal impacts may arise in downstream applications that may be affected by general progress in geometric machine learning, such as social network analysis and recommender systems. These two applications are known to have negative societal impacts in certain circumstances, so care must be taken in future related work to avoid major negative consequences.

## A.4  Edge Representations and Link Prediction

### A.4.1  Sign Invariant Link Prediction Decoders

Here, we present an ansatz for universal permutation invariant and sign invariant functions for $n = 2$, that is $f : \mathbb{R}^{2 \times k} \to \mathbb{R}^{d_{\text{out}}}$. Note that SignNet is only known to be universal for such functions for $n = 1$, where there are no permutation symmetries [Lim et al., 2023].

We will parameterize such functions as

$$f(v_1, \ldots v_k) = \varphi\left(v_1 \odot v_1, v_1 \odot \text{rev}(v_1), \ldots, v_k \odot v_k, v_k \odot \text{rev}(v_k)\right). \tag{9}$$

Here, $\text{rev} : \mathbb{R}^2 \to \mathbb{R}^2$ reverses the vector, so $\text{rev}(a)_1 = a_2$ and $\text{rev}(a)_2 = a_1$. Moreover, $\varphi : \mathbb{R}^{2 \times 2k} \to \mathbb{R}^{d_{\text{out}}}$ is a permutation invariant neural network, so $\varphi(PX) = \varphi(X)$ for all $2 \times 2$ permutation matrices $P$. Note that it is easy to parameterize permutation invariant functions $\varphi$ in a maximally expressive way, e.g. via DeepSets [Zaheer et al., 2017]. Now, we show that this parameterization is universal:

**Proposition 4.** *Functions $f : \mathbb{R}^{2 \times k} \to \mathbb{R}^{d_{\text{out}}}$ of the above form are permutation invariant and sign invariant, and they universally approximate permutation invariant and sign invariant functions.*

*Proof.* Invariance of $f$ is easy to see; let $P$ be a $2 \times 2$ permutation matrix and $s_i \in \{-1, 1\}$ for each $i$. Then

$$f(Pv_1 s_1, \ldots, Pv_k s_k) = \varphi\left((Pv_1 s_1) \odot (Pv_1 s_1), (Pv_1 s_1) \odot \text{rev}(Pv_1 s_1), \ldots\right) \tag{10}$$

$$= \varphi\left(P(v_1 s_1 \odot v_1 s_1), P(v_1 s_1 \odot \text{rev}(v_1 s_1)), \ldots\right) \tag{11}$$

$$= \varphi\left(P(v_1 \odot v_1), P(v_1 \odot \text{rev}(v_1)), \ldots\right) \tag{12}$$

$$= \varphi\left(v_1 \odot v_1, v_1 \odot \text{rev}(v_1), \ldots\right) \tag{13}$$

$$= f(v_1, \ldots, v_k), \tag{14}$$

where the second to last inequality is by permutation invariance of $\varphi$. Next, we show universal approximation.

Let $h : \mathbb{R}^{2 \times k} \to \mathbb{R}^{d_{\text{out}}}$ be a continuous permutation invariant and sign invariant function. Then by the decomposition theorem in Lim et al. [2023], we can write

$$h(v_1, \ldots, v_k) = \rho(\phi(v_1 v_1^\top), \ldots, \phi(v_k v_k^\top)), \tag{15}$$

for continuous functions $\rho$ and $\phi$. As a composition of continuous functions, the function $\psi : B \subseteq \mathbb{R}^{2 \times 2k} \to \mathbb{R}^{d_{\text{out}}}$ given by $\psi(A_1, \ldots, A_k) = \rho(\phi(A_1), \ldots, \phi(A_k))$ is continuous, where $B$ is the subset of $\mathbb{R}^{2 \times 2k}$ consisting of $(v_1 v_1^\top, \ldots, v_k v_k^\top)$ such that each $v_i \in \mathbb{R}^2$. Note that $\psi$ is permutation invariant on $B$, in the sense that for any $2 \times 2$ permutation matrix $P$, we have

$$\psi(PA_1 P^\top, \ldots, PA_k P^\top) = \psi(A_1, \ldots, A_k), \tag{16}$$

because if $v_i v_i^\top = A_i$, then

$$\psi(PA_1 P^\top, \ldots, PA_k P^\top) = h(Pv_1, \ldots, Pv_k) = h(v_1, \ldots, v_k) = \psi(A_1, \ldots, A_k), \tag{17}$$

by permutation invariance of $h$.

Now, we define our permutation invariant function $\varphi : C \subseteq \mathbb{R}^{2 \times 2k} \to \mathbb{R}^{d_{\text{out}}}$, on the domain

$$C = \{[v_1 \odot v_1, v_1 \odot \text{rev}(v_1), \ldots, v_k \odot v_k, v_k \odot \text{rev}(v_k)] : v_i \in \mathbb{R}^2\}. \tag{18}$$

We define $\varphi$ by

$$\varphi(A) = \psi \left( \begin{bmatrix} A_{1,1} & A_{2,2} \\ A_{2,2} & A_{2,1} \end{bmatrix}, \begin{bmatrix} A_{1,3} & A_{2,4} \\ A_{2,4} & A_{2,3} \end{bmatrix}, \ldots, \begin{bmatrix} A_{1,2k-1} & A_{2,2k} \\ A_{2,2k} & A_{2,2k-1} \end{bmatrix} \right). \tag{19}$$

To see that $\varphi$ is permutation invariant, we need only consider the case where $P = \begin{bmatrix} 0 & 1 \\ 1 & 0 \end{bmatrix}$, in which case

$$\varphi(PA) = \psi \left( \begin{bmatrix} A_{2,1} & A_{1,2} \\ A_{1,2} & A_{1,1} \end{bmatrix}, \begin{bmatrix} A_{2,3} & A_{1,4} \\ A_{1,4} & A_{1,3} \end{bmatrix}, \ldots, \begin{bmatrix} A_{2,2k-1} & A_{1,2k} \\ A_{1,2k} & A_{1,2k-1} \end{bmatrix} \right) \tag{20}$$

$$= \psi \left( P \begin{bmatrix} A_{1,1} & A_{2,2} \\ A_{2,2} & A_{2,1} \end{bmatrix} P^\top, P \begin{bmatrix} A_{1,3} & A_{2,4} \\ A_{2,4} & A_{2,3} \end{bmatrix} P^\top, \ldots, P \begin{bmatrix} A_{1,2k-1} & A_{2,2k} \\ A_{2,2k} & A_{2,2k-1} \end{bmatrix} P^\top \right) \tag{21}$$

$$= \psi \left( \begin{bmatrix} A_{1,1} & A_{2,2} \\ A_{2,2} & A_{2,1} \end{bmatrix}, \begin{bmatrix} A_{1,3} & A_{2,4} \\ A_{2,4} & A_{2,3} \end{bmatrix}, \ldots, \begin{bmatrix} A_{1,2k-1} & A_{2,2k} \\ A_{2,2k} & A_{2,2k-1} \end{bmatrix} \right) \quad (\psi \text{ perm. inv.}) \tag{22}$$

$$= \varphi(A), \tag{23}$$

where in the second equality, we use the fact that $A_{2,2j} = A_{1,2j}, j = 1, \ldots, k$ for $A \in C$, because $A_{2,2j} = (v_j \odot \text{rev}(v_j))_2 = (v_j \odot \text{rev}(v_j))_1 = A_{1,2j}$ for some $v_j \in \mathbb{R}^2$. Moreover, $\varphi$ is clearly continuous and sign invariant. Defining $f : \mathbb{R}^{2 \times k} \to \mathbb{R}^{d_{\text{out}}}$ using this $\varphi$, we compute that

$$f(v_1, \ldots v_k) = \varphi(v_1 \odot v_1, v_1 \odot \text{rev}(v_1), \ldots, v_k \odot v_k, v_k \odot \text{rev}(v_k)) \tag{24}$$

$$= \psi \left( \begin{bmatrix} v_{1,1}^2 & v_{1,1} v_{1,2} \\ v_{1,1} v_{1,2} & v_{1,2}^2 \end{bmatrix}, \ldots, \begin{bmatrix} v_{k,1}^2 & v_{k,1} v_{k,2} \\ v_{k,1} v_{k,2} & v_{k,2}^2 \end{bmatrix} \right) \tag{25}$$

$$= \psi \left( v_1 v_1^\top, \ldots, v_k v_k^\top \right) \tag{26}$$

$$= h(v_1, \ldots, v_k), \tag{27}$$

so we are done.

If $\varphi$ instead comes from a universally approximating class of permutation invariant neural networks (rather than being an arbitrary continuous permutation invariant function), then on a compact domain we can get $\epsilon$ approximation of $f$ to $h$ by letting $\varphi$ approximate $\psi$ to $\epsilon$ accuracy. $\qquad \square$

### A.4.2   Proof of Proposition 1

**Proposition 1.** *Let $f : \mathbb{R}^{n \times k} \to \mathbb{R}^{n \times d_{\text{out}}}$ be a permutation equivariant function, and let $V = [v_1, \ldots, v_k] \in \mathbb{R}^{n \times k}$ be $k$ orthonormal eigenvectors of an adjacency matrix $A$. Let nodes $i$ and $j$ be automorphic, and let $z_i$ and $z_j \in \mathbb{R}^{d_{\text{out}}}$ be their embeddings, i.e, the $i$th and $j$th row of $Z = f(V)$.*

- *If $f$ is sign invariant and the eigenvalues associated with the $v_l$ are simple and distinct, then $z_i = z_j$.*

- *If $f$ is basis invariant and $v_1, \ldots, v_k$ are a basis for some number of eigenspaces of $A$ then $z_i = z_j$.*

*Proof.* We only prove the basis invariance claim, as the sign invariance claim is a special case; basis invariance is sign invariance when eigenvalues are distinct.

Let $P \in \mathbb{R}^{n \times n}$ be a permutation matrix associated to an automorphism that maps node $i$ to node $j$, so $PAP^\top = A$ and $Pe_i = e_j$, where $e_l$ is the $l$th standard basis vector. Let $V_t = [v_{r_1}, \ldots, v_{r_{d_t}}]$ be the matrix whose columns are the eigenvectors $v_{r_l}$ that are associated to eigenvalue $\lambda_i$. The columns of $V_t$ are thus an orthonormal basis for the eigenspace associated to $\lambda_t$. Note that for any of these eigenvectors, we have

$$A(Pv_{r_l}) = PAP^\top(Pv_{r_l}) = PAv_{r_l} = P\lambda_i v_{r_l} = \lambda_t(Pv_{r_l}), \tag{28}$$

so $Pv_{r_l}$ is also an eigenvector of $A$ with eigenvalue $\lambda_t$. As $P$ is orthogonal, note that $Pv_{r_1}, \ldots, Pv_{r_{d_t}}$ is still an orthonormal basis of the eigenspace. Thus, there exists an orthogonal matrix $Q_t \in \mathbb{R}^{d_t \times d_t}$ such that $PV_t = V_t Q_t$—see Lim et al. [2023].

Repeat the above argument to get such a $Q_t$ for each of the eigenbases $V_1, \ldots, V_l$. We can then see that

$$
\begin{aligned}
z_j &= f(V_1, \ldots, V_l)_{j,:} \\
&= f(V_1 Q_1, \ldots, V_l Q_l)_{j,:} && \text{basis invariance} \\
&= f(PV_1, \ldots, PV_l)_{j,:} && \text{choice of } Q_t \\
&= (Pf(V_1, \ldots, V_l))_{j,:} && \text{permutation equivariance} \\
&= f(V_1, \ldots, V_l)_{i,:} && \text{choice of } P \\
&= z_i.
\end{aligned}
$$

So we are done. $\qquad\square$

### A.5 Sign Invariance and Structural Node or Node-Pair Encodings

In this section, we show that when the eigenvalues $\lambda_1, \ldots, \lambda_k$ are distinct, then sign invariant functions of the orthonormal eigenvectors $v_1, \ldots, v_k$ give structural node or node-pair representations. This can also be generalized in a straightforward way to larger tuples of nodes beyond pairs, though we only consider nodes and node-pairs for ease of exposition. First, we give formal definitions.

**Definition 1** (Structural Representations [Srinivasan and Ribeiro, 2019]). *Let $A \in \mathbb{R}^{n \times n}$ be the adjacency matrix of a graph on node set $\{1, \ldots, n\}$.*

*A function $f : \mathbb{R}^{n \times n} \to \mathbb{R}^n$ is a node structural representation if $f(PAP^\top) = Pf(A)$ for all $n \times n$ permutation matrices $P$.*

*A function $f : \mathbb{R}^{n \times n} \to \mathbb{R}^{n \times n}$ is a node-pair structural representation if $f(PAP^\top) = Pf(A)P^\top$ for all $n \times n$ permutation matrices $P$.*

Importantly, these structural representations are permutation equivariant functions of adjacency matrices, not arbitrary matrices. For each adjacency matrix $A$, let $V(A) = [v_1(A), \ldots, v_k(A)]$ be a choice of orthonormal eigenvectors for the first $k$ eigenvalues $\lambda_1(A), \ldots, \lambda_k(A)$. We assume in this section that these first $k$ eigenvalues are distinct for all $A$ under consideration, so $V(A)$ is defined up to sign flips. Let $h : \mathbb{R}^{n \times k} \to \mathbb{R}^n$ be a permutation equivariant function of sets, so $h(PX) = Ph(X)$ for all permutations matrices $P$. Then of course $h(PV(A)) = Ph(V(A))$, but this does not make $h$ a node structural encoding. This is because $A \mapsto h(V(A))$ is in general not a well-defined function of the adjacency, since the choice of $V(A)$ is not well-defined (the choices of sign are arbitrary). If we constrain $h$ to not depend on the signs (sign invariance), or to depend on the signs in a predictable way (sign equivariance), then we can compute structural node or node-pair encodings from eigenvectors.

We capture these observations in the below proposition. First, we define three types of functions:

- Let $f_{\text{node}} : \mathbb{R}^{n \times k} \to \mathbb{R}^n$ be sign invariant and permutation equivariant; that is, $f_{\text{node}}(Pv_1 s_1, \ldots, Pv_k s_k) = Pf_{\text{node}}(v_1, \ldots, v_k)$ for $s_i \in \{-1, 1\}$ and $P$ a permutation matrix.

- Let $f_{\text{decode}} : \mathbb{R}^{2 \times k} \to \mathbb{R}$ be sign invariant; that is, $f_{\text{decode}}(Sz_i, Sz_j) = f_{\text{decode}}(z_i, z_j)$ for $S \in \text{diag}(\{-1, 1\}^k)$.

- Let $f_{\text{equiv}} : \mathbb{R}^{n \times k} \to \mathbb{R}^{n \times k}$ be a permutation equivariant and sign equivariant function; that is, $f_{\text{equiv}}(PV(A)S) = Pf_{\text{equiv}}(V(A))S$ for $S \in \text{diag}(\{-1, 1\}^k)$ and $P$ a permutation matrix.

**Proposition 5.** *Let $\mathcal{A} \subseteq \mathbb{R}^{n \times n}$ denote the matrices with distinct first-$k$ eigenvalues. For $A \in \mathcal{A}$, let $V(A) = [v_1(A), \ldots, v_k(A)]$ be a choice of orthonormal eigenvectors of $A$, associated to the first-$k$ (distinct) eigenvalues $\lambda_1(A), \ldots, \lambda_k(A)$. Then*

*(a) The map $q_{\text{node}} : \mathcal{A} \to \mathbb{R}^n$ given by $q_{\text{node}}(A)_i = f_{\text{node}} \left( f_{\text{equiv}}(V(A)) \right)_i$ is well-defined and gives a structural node representation.*

*(b) The map $q_{\text{pair}} : \mathcal{A} \to \mathbb{R}^{n \times n}$ defined by $q_{\text{pair}}(A)_{i,j} = f_{\text{decode}} \left( f_{\text{equiv}}(V(A))_{i,:}, f_{\text{equiv}}(V(A))_{j,:} \right)$ is well-defined and gives a structural node-pair representation.*

Note that the identity mapping $V(A) \mapsto V(A)$ is permutation equivariant and sign equivariant, so using $f_{\text{node}}$ or $f_{\text{decode}}$ directly on eigenvectors also gives structural representations. The statement

(b) means that our link prediction pipeline with sign equivariant node features and sign invariant decoding produces structural node-pair representations.

*Proof.* **Part (a)** We first show that $q_{\text{node}} : \mathcal{A} \to \mathbb{R}^n$ is well-defined. Suppose we had another choice of eigenvectors, so the eigenvectors we input are $V(A)S$ for some $S \in \text{diag}(\{-1, 1\}^k)$. Then

$$f_{\text{node}}\left(f_{\text{equiv}}(V(A)S)\right) = f_{\text{node}}\left(f_{\text{equiv}}(V(A))S\right) = f_{\text{node}}\left(f_{\text{equiv}}(V(A))\right), \qquad (29)$$

where the first equality is by sign equivariance, and the second equality by sign invariance. Thus, the value of $q_{\text{node}}(A)$ is unchanged.

Now, let $P$ be any permutation matrix. Then for each eigenvector $v_i(A)$, $i \in [k]$, we have $(PAP^\top)Pv_i(A) = PAv_i(A) = \lambda_i(A)Pv_i(A)$, so $Pv_i(A)$ is an eigenvector of $PAP^\top$ associated to $\lambda_i(A) = \lambda_i(PAP^\top)$. Hence, we denote $v_i(PAP^\top) = Pv_i(A)$ (the choice of sign does not matter as $q$ does not depend on the sign. Now, we have that

$$q_{\text{node}}(PAP^\top) = f_{\text{node}}\left(f_{\text{equiv}}(V(PAP^\top))\right) \qquad (30)$$
$$= f_{\text{node}}\left(f_{\text{equiv}}(PV(A))\right) \qquad (31)$$
$$= Pf_{\text{node}}\left(f_{\text{equiv}}(V(A))\right) \qquad (32)$$
$$= Pq_{\text{node}}(A) \qquad (33)$$

where the second to last equality is by permutation equivariance of $f_{\text{node}}$ and $f_{\text{equiv}}$.

**Part (b)** That $q_{\text{pair}} : \mathcal{A} \to \mathbb{R}^{n \times n}$ is well-defined follows from a similar argument to the $q_{\text{node}}$ case. Let $P$ be a permutation matrix, and $\sigma : [n] \to [n]$ its underlying permutation. We compute that

$$q_{\text{pair}}(PAP^\top)_{i,j} = f_{\text{decode}}\left(f_{\text{equiv}}(V(PAP^\top))_{i,:}, f_{\text{equiv}}(V(PAP^\top))_{j,:}\right) \qquad (34)$$
$$= f_{\text{decode}}\left(f_{\text{equiv}}(PV(A))_{i,:}, f_{\text{equiv}}(PV(A))_{j,:}\right) \qquad (35)$$
$$= f_{\text{decode}}\left([Pf_{\text{equiv}}(V(A))]_{i,:}, [Pf_{\text{equiv}}(V(A))]_{j,:}\right) \qquad (36)$$
$$= f_{\text{decode}}\left(f_{\text{equiv}}(V(A))_{\sigma^{-1}(i),:}, f_{\text{equiv}}(V(A))_{\sigma^{-1}(j),:}\right) \qquad (37)$$
$$= q_{\text{pair}}(A)_{\sigma^{-1}(i),\sigma^{-1}(j)} \qquad (38)$$
$$= (Pq_{\text{pair}}(A)P^\top)_{i,j} \qquad (39)$$

$\square$

### A.5.1 Sign Equivariance is Provably More Expressive for Link Prediction

Our arguments in Section 2.1 and Figure 2 explain why we can expect sign equivariant models to be more powerful than sign invariant models in link prediction. To give a theoretically rigorous explanation, here we provide an example where sign equivariant models can provably compute more expressive link representations than sign invariant models.

Consider a cycle graph $C_{2k}$ for some even length $2k$, where $k \geq 3$. All nodes are automorphic in this graph, so any model based on structural node representations must assign the same representation to each node-pair. For instance, consider the eigenvalue $-2$ of the adjacency matrix, which is a simple eigenvalue with eigenvector $[1, -1, 1, -1, \ldots, 1, -1]$ [Lee et al., 1992]. Then a sign invariant model will lose the sign information and map each node to the same encoding, which means that each node-pair will also have the same encoding. However, a sign equivariant model can preserve the sign of each node (for instance by learning the identity function). Then for any pair of nodes that are one hop away, it can take a dot product to compute the pair representation $-1$, whereas it can take a dot product between any nodes that are two hops away to compute the pair representation $1$. Of course, using more eigenvectors would allow for more complex representations to be computed.

### A.5.2 More on Sign Equivariance and Link Prediction

Key to our method is the ability to update a positional node embedding in an equivariant way, which respects the graph symmetries. To elaborate, consider the aforementioned definition of node positional encodings as samples from a permutation equivariant probability distribution over node features [Srinivasan and Ribeiro, 2019]. Laplacian eigenvector positional embeddings are samples from the distribution of orthonormal bases of the eigenspaces of the Laplacian. Our sign equivariance

based approach is possible because the randomness in Laplacian eigenvector positional encodings is exceptionally structured (consisting only of sign flips when eigenvalues are distinct). In contrast, a general way to obtain structural pair representations from node positional embeddings is to average some function over the randomness of the positional encoding (i.e., over many samples of the positional encoding) [Srinivasan and Ribeiro, 2019], but this is highly expensive, often intractable, and introduces substantial variance into the learning procedure. For instance, one may have to average samples of the $n!$ assignments of unique node identifiers [Murphy et al., 2019] or approximate an integral over Gaussian random features [Abboud et al., 2021].

### A.6 Proof of Proposition 2, Orthogonal Equivariance via Sign Equivariance

**Proposition 2.** *Consider a domain $\mathcal{X} \subseteq \mathbb{R}^{n \times d}$ such that each $X \in \mathcal{X}$ has distinct covariance eigenvalues, and let $R_X$ be a choice of orthonormal eigenvectors of $\mathrm{cov}(X)$ for each $X \in \mathcal{X}$. If $h : \mathcal{X} \subseteq \mathbb{R}^{n \times d} \to \mathbb{R}^{n \times d}$ is sign equivariant, and if $f(X) = h(XR_X)R_X^\top$, then $f$ is well defined and orthogonally equivariant.*

*Moreover, is $h$ is from a universal class of sign equivariant functions, then the $f$ of the above form universally approximate $O(k)$ equivariant functions on $\mathcal{X}$.*

*Proof.* First, we show that $f$ is well defined. $R_X$ is only unique up to sign flips, as $R_X S$ is an orthonormal set of eigenvectors of $\mathrm{cov}(X)$ for $S \in \mathrm{diag}(\{-1, 1\}^k)$. However, no matter the choice of signs, $f(X)$ takes the same value, since

$$h(XR_X S)(R_X S)^\top = h(XR_X S)S^\top R_X^\top \tag{40}$$

$$= h(XR_X)SS^\top R_X^\top \qquad \text{sign equivariance} \tag{41}$$

$$= h(XR_X)R_X^\top. \tag{42}$$

Next, we show that $f$ is $O(k)$ equivariant. Let $Q \in O(k)$ be any orthogonal matrix. Note that

$$\mathrm{cov}(XQ) = \left(XQ - \frac{1}{n}\mathbf{1}\mathbf{1}^\top XQ\right)^\top \left(XQ - \frac{1}{n}\mathbf{1}\mathbf{1}^\top XQ\right) = Q^\top \mathrm{cov}(X)Q. \tag{43}$$

Thus, $Q^\top R_X$ is an orthonormal set of eigenvectors of $\mathrm{cov}(XQ)$. This means that there is a choice of signs $S \in \mathrm{diag}(\{-1, 1\}^k)$ such that $Q^\top R_X S = R_{XQ}$. Hence, we have that

$$f(XQ) = h(XQR_{XQ})R_{XQ}^\top \tag{44}$$

$$= h(XQQ^\top R_X S)(Q^\top R_X S)^\top \tag{45}$$

$$= h(XR_X)SS^\top R_X^\top Q \qquad \text{sign equivariance} \tag{46}$$

$$= h(XR_X)R_X^\top Q \tag{47}$$

$$= f(X)Q^\top, \tag{48}$$

so $f$ is $O(k)$ equivariant.

**Universal Approximation.** Our proof of the universality of this class of functions builds on the proof of the universality of frame averaging [Puny et al., 2022]. Let $f_{\text{target}}$ be a continuous $O(k)$ equivariant function and let $\epsilon > 0$ be a desired approximation accuracy. Then $f_{\text{target}}$ is also sign equivariant (as the sign matrices $S \in \mathrm{diag}(\{-1, 1\}^k)$ are orthogonal).

Hence, by sign equivariant universality, we can choose a sign equivariant $h$ such that $\|h(X) - f_{\text{target}}(X)\| < \epsilon$ for all $X \in \mathcal{X}$ (where $\|\cdot\|$ is the Frobenius norm). Define the $O(k)$ equivariant $f(X) = h(XR_X)R_X^\top$. Then for all $X \in \mathcal{X}$ we have that

$$\|f_{\text{target}}(X) - f(X)\| = \|f_{\text{target}}(X) - h(XR_X)R_X^\top\| \tag{49}$$

$$= \|f_{\text{target}}(X)R_X R_X^\top - h(XR_X)R_X^\top\| \qquad R_X \text{ orthogonal} \tag{50}$$

$$= \|f_{\text{target}}(XR_X)R_X^\top - h(XR_X)R_X^\top\| \quad \text{orthogonal equivariance} \tag{51}$$

$$= \|f_{\text{target}}(XR_X) - h(XR_X)\| \qquad R_X \text{ orthogonal} \tag{52}$$

$$< \epsilon. \tag{53}$$

So $f$ approximates $f_{\text{target}}$ within $\epsilon$ accuracy on $\mathcal{X}$, and we are done. $\qquad\square$

## B Sign Equivariant Linear Maps

### B.1 Sign Equivariant Linear Map Characterization

We first prove our result characterizing the form of the equivariant linear maps from $\mathbb{R}^{n \times k} \to \mathbb{R}^{n' \times k}$.

**Lemma 1.** *A linear map $W : \mathbb{R}^{n \times k} \to \mathbb{R}^{n' \times k}$ is sign equivariant if and only if it can be written as*

$$W(X) = [W_1 X_1 \ \ldots \ W_k X_k] \tag{54}$$

*for some linear maps $W_1, \ldots, W_k : \mathbb{R}^n \to \mathbb{R}^{n'}$, where $X_i \in \mathbb{R}^n$ is the ith column of $X \in \mathbb{R}^{n \times k}$.*

*Proof.* For one direction, suppose $W$ can be written as in equation 4. To see that $W$ is sign equivariant, note that for any $S \in \mathrm{diag}(\{-1, 1\}^k)$, we have

$$W(XS) = [s_1 W_1 X_1 \ \ldots \ s_k W_k X_k] = [W_1 X_1 \ \ldots \ W_k X_k] S = W(X)S. \tag{55}$$

For the other direction, let $W$ be a sign equivariant linear map. For any $i' \in [n']$ and $j' \in [k]$, we can write the action of $W$ as

$$W(X)_{i',j'} = \sum_{i=1}^{n} \sum_{j=1}^{k} W_{i',j'}^{i,j} X_{i,j}, \tag{56}$$

where $W_{i',j'}^{i,j} \in \mathbb{R}$ are coefficients representing the linear map. Let $c \neq j'$ be a column that is not $j'$. Further, for any row $l \in [n]$, let $\tilde{X} \in \mathbb{R}^{n \times k}$ be such that $\tilde{X}_{l,c} = 1$, and $\tilde{X}$ is zero elsewhere. Then we have that

$$W(\tilde{X})_{i',j'} = W_{i',j'}^{l,c}. \tag{57}$$

Now, let $S \in \mathrm{diag}(\{-1, 1\}^k)$ have a $-1$ in the $j'$th column and a $1$ elsewhere. Then $\tilde{X}S = \tilde{X}$. This implies that

$$W_{i',j'}^{l,c} = W(\tilde{X})_{i',j'} \tag{58}$$

$$= W(\tilde{X}S)_{i',j'} \tag{59}$$

$$= -W(\tilde{X})_{i',j'} \tag{60}$$

$$= -W_{i',j'}^{l,c}, \tag{61}$$

where in the second to last equality we used sign equivariance. This implies that $W_{i',j'}^{l,c} = 0$.

Hence, for any $i' \in [n'], j' \in [k']$, we have that $W(X)_{i',j'}$ only depends on $X_{j'}$, so we are done. $\square$

### B.2 Sign Equivariant Linear Maps between Tensor Representations

Since the sign equivariant linear maps from $\mathbb{R}^{n \times k} \to \mathbb{R}^{n' \times k}$ are very weak, we now characterize sign equivariant linear maps between higher order tensor representations, as past work has done for other groups [Maron et al., 2018, 2019, Finzi et al., 2021]. In particular, we will consider representations $\mathbb{R}^{k^m}$ for natural numbers $m$. The action of $s \in \{-1, 1\}^k$ on $V \in \mathbb{R}^{k^m}$ is as follows:

$$(s \cdot V)_{i_1,\ldots,i_m} = s_{i_1} \cdots s_{i_m} V_{i_1,\ldots,i_m}, \tag{62}$$

for $i_1, \ldots, i_m \in [k]$. We now prove a result showing that there are no sign equivariant linear maps between many pairs of tensor representations.

**Proposition 6.** *If $m_1 + m_2$ is odd, then the only sign equivariant linear map $L : \mathbb{R}^{k^{m_1}} \to \mathbb{R}^{k^{m_2}}$ is the zero map.*

*Proof.* Let $L : \mathbb{R}^{k^{m_2} \times k^{m_1}}$ be the matrix associated with a sign equivariant linear map from $\mathbb{R}^{k^{m_1}} \to \mathbb{R}^{k^{m_2}}$. This means that for $V \in \mathbb{R}^{k^{m_1}}$, for $i_1, \ldots, i_{m_2} \in [k]$ we have that

$$(LV)_{i_1,\ldots,i_{m_2}} = \sum_{j_1,\ldots,j_{m_1}=1}^{k} L_{i_1,\ldots,i_{m_2},j_1,\ldots,j_{m_1}} V_{j_1,\ldots,j_{m_1}}. \tag{63}$$

Then by sign equivariance, we have that

$$L(s \cdot V) = s \cdot (LV), \tag{64}$$

for $s \in \{-1, 1\}^k$, which means that for all $i_1, \ldots, i_{m_2} \in [k]$,

$$\sum_{j_1, \ldots, j_{m_1} = 1}^{k} L_{i_1, \ldots, i_{m_2}, j_1, \ldots, j_{m_1}} s_{j_1} \cdots s_{j_{m_1}} V_{j_1, \ldots, j_{m_1}} \tag{65}$$

$$= s_{i_1} \cdots s_{i_{m_2}} \sum_{j_1, \ldots, j_{m_1} = 1}^{k} L_{i_1, \ldots, i_{m_2}, j_1, \ldots, j_{m_1}} V_{j_1, \ldots, j_{m_1}}, \tag{66}$$

which means that

$$\sum_{j_1, \ldots, j_{m_1} = 1}^{k} L_{i_1, \ldots, i_{m_2}, j_1, \ldots, j_{m_1}} s_{i_1} \cdots s_{i_{m_2}} s_{j_1} \cdots s_{j_{m_1}} V_{j_1, \ldots, j_{m_1}} \tag{67}$$

$$= \sum_{j_1, \ldots, j_{m_1} = 1}^{k} L_{i_1, \ldots, i_{m_2}, j_1, \ldots, j_{m_1}} V_{j_1, \ldots, j_{m_1}}. \tag{68}$$

This gives that $s \cdot L = L$. Choose some arbitrary indices $i_1, \ldots, i_{m_2}, j_1, \ldots, j_{m_1} \in [k]$. Since $m_1 + m_2$ is odd, then one of these values appears an odd number of times — say it is some index $p \in [k]$. Let $s \in \{-1, 1\}^k$ have $s_i = 1$ for $i \neq p$ and $s_p = -1$. Then $s \cdot L = L$ implies that

$$-L_{i_1, \ldots, i_{m_2}, j_1, \ldots, j_{m_1}} = L_{i_1, \ldots, i_{m_2}, j_1, \ldots, j_{m_1}}. \tag{69}$$

Which implies that $L$ is zero at these indices, and hence $L = 0$ everywhere, since these were arbitrarily chosen indices. $\qquad\square$

This implies that we cannot map from $\mathbb{R}^k \to \mathbb{R}^{k^2}$ using sign equivariant linear maps. Thus, if we want to lift our order-one tensor input to higher tensor orders in a linear way, then we need to at least map to third order tensors in $\mathbb{R}^{k^3}$. This requires substantial memory cost. Furthermore, we next show that even if we could map to third order tensors, learning representations of third order tensors with sign equivariant linear maps is expensive.

**Proposition 7.** *The dimension of the space of sign equivariant linear maps from $\mathbb{R}^{k^{m_1}} \to \mathbb{R}^{k^{m_2}}$ is*

$$\frac{1}{2^k} \sum_{s \in \{-1, 1\}^k} (s_1 + \ldots + s_k)^{m_1 + m_2}. \tag{70}$$

*Proof.* This is a direct consequence of the First Projection Formula: see Fulton and Harris [2013] Section 2.2. In particular, it can be seen that the dimension of this space is equal to

$$\frac{1}{2^k} \sum_{S \in \text{Diag}(\{-1, 1\}^k)} \text{trace}(S^{\otimes(m_1 + m_2)}) \tag{71}$$

by the First Projection Formula. Since $\text{trace}(A \otimes B) = \text{trace}(A) \cdot \text{trace}(B)$, this is equal to

$$\frac{1}{2^k} \sum_{S \in \text{Diag}(\{-1, 1\}^k)} \text{trace}(S)^{m_1 + m_2} = \frac{1}{2^k} \sum_{s \in \{-1, 1\}^k} (s_1 + \ldots + s_k)^{m_1 + m_2}. \tag{72}$$

$\qquad\square$

Using Proposition 7, we have numerically computed the dimension of the space of sign equivariant linear maps from $\mathbb{R}^{k^3} \to \mathbb{R}^{k^3}$; see Table 4. The dimension appears to be $15k^3 - 30k^2 + 16k$ for $k$ up to 20. In particular, when $k = 8$, we compute that the space of sign equivariant linear maps from $\mathbb{R}^{k^3} \to \mathbb{R}^{k^3}$ is of dimension 5888, which is already quite large.

Table 4: Dimension of the space of sign equivariant linear maps between third-order tensor representations $\mathbb{R}^{k^3} \to \mathbb{R}^{k^3}$.

| $k$ | Dimension |
|---|---|
| 1 | 1 |
| 2 | 32 |
| 3 | 183 |
| 4 | 544 |
| 5 | 1,205 |
| 6 | 2,256 |
| 7 | 3,787 |
| 8 | 5,888 |
| 9 | 8,649 |
| 10 | 12,160 |
| 11 | 16,511 |
| 12 | 21,792 |
| 13 | 28,093 |
| 14 | 35,504 |
| 15 | 44,115 |
| 16 | 54,016 |
| 17 | 65,297 |
| 18 | 78,048 |
| 19 | 92,359 |
| 20 | 108,320 |

## C   Characterization of Sign Equivariant Polynomials

In this Appendix, we characterize the form of the sign equivariant polynomials. This is useful, because for a finite group, equivariant polynomials universally approximate equivariant continuous functions [Yarotsky, 2022]; thus, if a model universally approximates equivariant polynomials, then it universally approximates equivariant continuous functions. Using equivariant polynomials to analyze or develop equivariant machine learning models has been done successfully in many contexts [Zaheer et al., 2017, Yarotsky, 2022, Segol and Lipman, 2019, Dym and Maron, 2021, Maron et al., 2019, 2020, Villar et al., 2021, Dym and Gortler, 2022, Puny et al., 2023].

### C.1   Sign Invariant Polynomials $\mathbb{R}^k \to \mathbb{R}$

Next, we characterize the form of sign invariant and equivariant polynomials. For simplicity, we start with the case of sign invariant polynomials $p : \mathbb{R}^k \to \mathbb{R}$. The sign equivariant polynomials take a very similar form. We can write any polynomial from $\mathbb{R}^k$ to $\mathbb{R}$ in the form

$$p(v) = \sum_{d_1,\dots,d_k=0}^{D} \mathbf{W}_{d_1,\dots,d_k} v_1^{d_1} \cdots v_k^{d_k} \tag{73}$$

for some coefficients $\mathbf{W}_{d_1,\dots,d_k} \in \mathbb{R}$ and some $D \in \mathbb{N}$. Sign invariance tells us that for any $S = \mathrm{diag}(s_1,\dots,s_k) \in \mathrm{diag}(\{-1,1\}^k)$, we must have

$$\sum_{d_1,\dots,d_k=0}^{D} \mathbf{W}_{d_1,\dots,d_k} v_1^{d_1} \cdots v_k^{d_k} = p(v) = p(Sv) = \sum_{d_1,\dots,d_k=0}^{D} \mathbf{W}_{d_1,\dots,d_k} s_1^{d_1} \cdots s_k^{d_k} v_1^{d_1} \cdots v_k^{d_k}. \tag{74}$$

This holds for any $v \in \mathbb{R}^k$, so for all choices of $d_1,\dots,d_k$ we must have

$$\mathbf{W}_{d_1,\dots,d_k} = s_1^{d_1} \cdots s_k^{d_k} \mathbf{W}_{d_1,\dots,d_k}, \quad \text{for all } (s_1,\dots,s_k) \in \{-1,1\}^k. \tag{75}$$

Note that $s_i^{d_i} = 1$ if $d_i$ is an even number. Hence, there are no constraints on $\mathbf{W}_{d_1,\dots,d_k}$ if all $d_i$ are even. On the other hand, suppose $d_j$ is odd for some $j$. Let $s_i = 1$ for $i \neq j$ and $s_j = -1$. Then the

constraint says that $\mathbf{W}_{d_1,\ldots,d_k} = -\mathbf{W}_{d_1,\ldots,d_k}$, so we must have $\mathbf{W}_{d_1,\ldots,d_k} = 0$. To summarize, we have

$$\mathbf{W}_{d_1,\ldots,d_k} = \begin{cases} \text{free} & d_i \text{ even for each } i \\ 0 & \text{else} \end{cases} \tag{76}$$

Where being free means that the coefficient may take any value in $\mathbb{R}$. Thus, any sign invariant $p$ only has terms where each variable $v_i$ is raised to an even power. It is also easy to see that any polynomial $p$ where each variable $v_i$ is raised to only even powers is sign invariant, so we have the following proposition:

**Proposition 8.** *A polynomial $p : \mathbb{R}^k \to \mathbb{R}$ is sign invariant if and only if it can be written*

$$p(v) = \sum_{d_1,\ldots,d_k=0}^{D} \mathbf{W}_{d_1,\ldots,d_k} v_1^{2d_1} \cdots v_k^{2d_k}, \tag{77}$$

*for some coefficients $\mathbf{W}_{d_1,\ldots,d_k} \in \mathbb{R}$ and $D \in \mathbb{N}$.*

*In other words, $p$ is sign invariant if and only if there exists a polynomial $q : \mathbb{R}^k \to \mathbb{R}$ such that $p(v) = q(v_1^2, \ldots, v_k^2)$.*

### C.2 Sign Equivariant Polynomials $\mathbb{R}^k \to \mathbb{R}^k$

The case of sign equivariant polynomials $p : \mathbb{R}^k \to \mathbb{R}^k$ is very similar. For $l \in [k]$, the $l$th output dimension of any polynomial $p : \mathbb{R}^k \to \mathbb{R}^k$ can be written

$$p(v)_l = \sum_{d_1,\ldots,d_k=0}^{D} \mathbf{W}_{d_1,\ldots,d_k}^{(l)} v_1^{d_1} \cdots v_k^{d_k}, \tag{78}$$

where $\mathbf{W}_{d_1,\ldots,d_k}^{(l)} \in \mathbb{R}$ are coefficients (note the extra $l$ index, so there are $k$ times more coefficients than in the invariant case). By sign equivariance, we have

$$s_l \cdot p(v)_l = p(Sv)_l \tag{79}$$

$$s_l \cdot \sum_{d_1,\ldots,d_k=0}^{D} \mathbf{W}_{d_1,\ldots,d_k}^{(l)} v_1^{d_1} \cdots v_k^{d_k} = \sum_{d_1,\ldots,d_k=0}^{D} \mathbf{W}_{d_1,\ldots,d_k}^{(l)} s_1^{d_1} \cdots s_k^{d_k} v_1^{d_1} \cdots v_k^{d_k}. \tag{80}$$

As this holds for all inputs $v \in \mathbb{R}^k$, we have the following constraints on the coefficients:

$$s_l \mathbf{W}_{d_1,\ldots,d_k}^{(l)} = s_1^{d_1} \cdots s_k^{d_k} \mathbf{W}_{d_1,\ldots,d_k}^{(l)} \tag{81}$$

$$\mathbf{W}_{d_1,\ldots,d_k}^{(l)} = s_l \cdot s_1^{d_1} \cdots s_k^{d_k} \mathbf{W}_{d_1,\ldots,d_k}^{(l)}, \tag{82}$$

where we use the fact that $s_l = 1/s_l$ since $s_l \in \{-1, 1\}$. If $d_j$ is odd for $j \neq l$, then similarly to the invariant case, we can take $s_i = 1$ for $i \neq j$ and $s_j = -1$ in the above equation to see that $\mathbf{W}_{d_1,\ldots,d_k}^{(l)} = 0$. If $d_l$ is even, then $d_l + 1$ is odd, so we have that $\mathbf{W}_{d_1,\ldots,d_k}^{(l)} = 0$ by the same argument. Thus, we must have

$$\mathbf{W}_{d_1,\ldots,d_k}^{(l)} = \begin{cases} \text{free} & d_l \text{ odd, and } d_i \text{ even for each } i \neq l \\ 0 & \text{else} \end{cases}. \tag{83}$$

Thus, the $l$th entry $p(v)_l$ only contains monomials of the term $v_1^{2d_1} \cdots v_l^{2d_l+1} \cdots v_k^{2d_k}$, where each term besides $v_l$ is raised to an even power. We can factor out a $v_l$ and write such terms as $v_l \cdot v_1^{2d_1} \cdots v_k^{2d_k}$. It is also easy to see that any polynomial with monomials only of this form is sign equivariant. Thus, we have proven Proposition 9.

**Proposition 9.** *A polynomial $p : \mathbb{R}^k \to \mathbb{R}^k$ is sign equivariant if and only if it can be written*

$$p(v)_l = v_l \cdot \left( \sum_{d_1,\ldots,d_k=0}^{D} \mathbf{W}_{d_1,\ldots,d_k}^{(l)} v_1^{2d_1} \cdots v_k^{2d_k} \right). \tag{84}$$

*In vector format, $p$ is sign equivariant if and only if it can be written as $p(v) = v \odot p_{\mathrm{inv}}(v)$ for a sign invariant polynomial $p_{\mathrm{inv}} : \mathbb{R}^k \to \mathbb{R}^k$.*

## C.3   Sign Equivariant Polynomials $\mathbb{R}^{n \times k} \to \mathbb{R}^{n' \times k}$

Finally, we will handle the case of polynomials $p : \mathbb{R}^{n \times k} \to \mathbb{R}^{n' \times k}$ equivariant to $\mathrm{diag}(\{-1, 1\}^k)$. This is the case we most often deal with in practice, when we have input $V = [v_1 \quad \ldots \quad v_k]$ for $k$ eigenvectors $v_i \in \mathbb{R}^n$ of some $n \times n$ matrix. For $a \in [n']$ and $b \in [k]$, the $(a, b)$th output of a polynomial $\mathbb{R}^{n \times k} \to \mathbb{R}^{n' \times k}$ is

$$p(V)_{a,b} = \sum_{d_{i,j}=0}^{D} \mathbf{W}_{\mathbf{d}}^{(a,b)} \prod_{i=1}^{n} \prod_{j=1}^{k} V_{i,j}^{d_{i,j}}, \tag{85}$$

where the sum ranges over $d_{i,j} \in \{0, \ldots, D\}$ for $i \in [n]$ and $j \in [k]$, and $\mathbf{d} = (d_{1,1}, \ldots, d_{n,1}, d_{1,2}, \ldots, d_{n,k})$ is a shorthand to index coefficients $\mathbf{W}_{\mathbf{d}}^{(a,b)} \in \mathbb{R}$. By sign equivariance, we have that:

$$s_b \cdot p(V)_{a,b} = p(VS)_{a,b} \tag{86}$$

$$s_b \cdot \sum_{d_{i,j}=0}^{D} \mathbf{W}_{\mathbf{d}}^{(a,b)} \prod_{i=1}^{n} \prod_{j=1}^{k} V_{i,j}^{d_{i,j}} = \sum_{d_{i,j}=0}^{D} \mathbf{W}_{\mathbf{d}}^{(a,b)} s_1^{\tilde{d}_1} \cdots s_k^{\tilde{d}_k} \prod_{i=1}^{n} \prod_{j=1}^{k} V_{i,j}^{d_{i,j}}, \tag{87}$$

where $\tilde{d}_j = \sum_{i'=1}^{n} d_{i',j}$ is the number of times that an entry from column $j$ of $V$ appears in the product $\prod_{i=1}^{n} \prod_{j=1}^{k} V_{i,j}^{d_{i,j}}$. As this holds over all $V$, we thus have that

$$\mathbf{W}_{\mathbf{d}}^{(a,b)} = s_b \cdot s_1^{\tilde{d}_1} \cdots s_k^{\tilde{d}_k} \cdot \mathbf{W}_{\mathbf{d}}^{(a,b)}. \tag{88}$$

By analogous arguments to the previous subsections, if $\tilde{d}_j$ is odd for $j \neq b$, we have that the $\mathbf{W}_{\mathbf{d}}^{(a,b)} = 0$. Likewise, if $\tilde{d}_b$ is even, we have $\mathbf{W}_{\mathbf{d}}^{(a,b)} = 0$. Thus, the constraint on $\mathbf{W}$ is

$$\mathbf{W}_{\mathbf{d}}^{(a,b)} = \begin{cases} \text{free} & \sum_i d_{i,b} \text{ odd, and } \sum_i d_{i,j} \text{ even for each } j \neq b \\ 0 & \text{else} \end{cases}. \tag{89}$$

In particular, this means that the only nonzero terms in the sum that defines $p(V)_{a,b}$ have an even number of entries from column $j$ for $j \neq b$, and an odd number of entries from column $b$. Thus, each term can be written as $V_{i_{\mathbf{d}}, b} \cdot p_{\mathrm{inv}}(V)_{\mathbf{d}}$ for some index $i_{\mathbf{d}} \in [n]$ and sign invariant polynomial $p_{\mathrm{inv}}$. Moreover, it can be seen that any polynomial that only has terms of this form is sign equivariant. Thus, we have shown the following proposition:

**Proposition 10.** *A polynomial $p : \mathbb{R}^{n \times k} \to \mathbb{R}^{n' \times k}$ is sign equivariant if and only if it can be written as*

$$p(V)_{a,b} = \sum_{d_{i,j}=0}^{D} \mathbf{W}_{\mathbf{d}}^{(a,b)} V_{i_{\mathbf{d}}, b} \cdot p_{\mathrm{inv}}(V)_{\mathbf{d}}, \tag{90}$$

*where $p_{\mathrm{inv}}$ is a sign invariant polynomial, the sum ranges over all $\mathbf{d}$, and $i_{\mathbf{d}} \in [n]$ for each $\mathbf{d}$.*

Now, we show that this implies Theorem 1. In particular, we will write $p$ in the form

$$p(V) = W^{(2)} \left( (W^{(1)} V) \odot q_{\mathrm{inv}}(V) \right), \tag{91}$$

for sign equivariant linear maps $W^{(2)}$ and $W^{(1)}$, and a sign equivariant polynomial $q_{\mathrm{inv}}$. To do so, let $\tilde{D}$ denote the number of all possible $\mathbf{d}$ that the sum in equation 90 ranges over. We take $W^{(1)} : \mathbb{R}^{n \times k} \to \mathbb{R}^{\tilde{D} n' \times k}$ and $W^{(2)} : \mathbb{R}^{\tilde{D} n' \times k} \to \mathbb{R}^{n' \times k}$. These sign equivariant linear maps have to act independently on each column of their input, so $W^{(1)} V = [W_1^{(1)} v_1, \ldots W_k^{(1)} v_k]$ for linear maps $W_i^{(1)} : \mathbb{R}^n \to \mathbb{R}^{\tilde{D} n'}$. We define $W_b^{(1)}$ to be the linear map such that $(W_b^{(1)} v_b)_{\mathbf{d}, a} = W_{\mathbf{d}}^{(a,b)} V_{i_{\mathbf{d}}, b}$ for $a \in [n']$. For the sign invariant polynomial $q_{\mathrm{inv}}$, we take $q_{\mathrm{inv}}(V)_{\mathbf{d}, a} = p_{\mathrm{inv}}(V)_{\mathbf{d}}$.

Finally, we define $W^{(2)}$ to compute the sum in equation 90. In particular, for $X = [x_1, \ldots, x_k] \in \mathbb{R}^{\tilde{D} n' \times k}$ we write $W^{(2)} X = [W_1^{(2)} x_1, \ldots, W_k^{(2)} x_k]$, where $(W_b^{(2)} x_b)_a = \sum_{\mathbf{d}} x_{i_{\mathbf{d}}, b}$. It can be seen that with these definitions of $W^{(2)}, W^{(1)}$, and $q_{\mathrm{inv}}$, we have written $p$ in the desired form.

## C.4 Sign Invariant Polynomials and SignNet

For completeness, here we state the form of the sign invariant polynomials $p : \mathbb{R}^{n \times k} \to \mathbb{R}$ on inputs $V = [v_1, \ldots, v_k] \in \mathbb{R}^{n \times k}$. The derivation very closely follows that of the sign equivariant polynomials from $\mathbb{R}^{n \times k} \to \mathbb{R}^{n' \times k}$ in Appendix C.3, so we omit this derivation.

**Proposition 11.** *A polynomial $p : \mathbb{R}^{n \times k} \to \mathbb{R}$ is sign invariant if and only if it can be written*

$$p(V) = \sum_{d_{i,j}=0}^{D} \mathbf{W_d} \prod_{i=1}^{n} \prod_{j=1}^{k} V_{i,j}^{d_{i,j}}, \tag{92}$$

*where $\mathbf{W_d} \neq 0$ for $\mathbf{d} = (d_{1,1}, \ldots, d_{n,1}, d_{1,2}, \ldots, d_{n,k})$ only if $\sum_{i=1}^{n} d_{i,j}$ is even for each column $j \in [k]$.*

*In particular, $p$ is sign invariant if and only if there is a polynomial $q : \mathbb{R}^{n \times n \times k} \to \mathbb{R}$ such that $p(V) = q([V_{i_1,j} \cdot V_{i_2,j}]_{i_1 \in [n], i_2 \in [n], j \in [k]})$.*

The polynomials $V \mapsto V_{i_1,j} \cdot V_{i_2,j}$ for $i_1, i_2 \in [n]$ and $j \in [k]$ are thus generators of the ring of sign invariant polynomials from $\mathbb{R}^{n \times k} \to \mathbb{R}$.

Notably, Lim et al. [2023] propose universal sign invariant neural architectures, but do not characterize or otherwise use the sign invariant polynomials. Instead, their proof of universality uses topological constructions and shows that all sign invariant continuous functions can be decomposed in a simple form—namely, $\rho([\phi(v_i) + \phi(-v_i)]_{i=1,\ldots,k})$ for continuous functions $\rho$ and $\phi$. Our characterization of sign invariant polynomials provides another path to developing and analyzing the expressive power of sign invariant architectures.

In particular, we can give an alternative proof for the universality of SignNet.

**Proposition 12** (Universality of SignNet). *Let $f : \mathcal{X} \subseteq \mathbb{R}^{n \times k} \to \mathbb{R}$ be a continuous sign invariant function on a compact domain $\mathcal{X}$, and let $\epsilon > 0$. Then there exists a continuous $\rho : \mathbb{R}^{n^2 k} \to \mathbb{R}$ and continuous $\phi : \mathbb{R}^n \to \mathbb{R}^{n^2}$ such that $|f(V) - \rho([\phi(v_i) + \phi(-v_i)]_{i=1,\ldots,k})| < \epsilon$ for all $V \in \mathcal{X}$.*

*Proof.* First, let $p$ be a sign invariant polynomial that approximates $f$ to within $\epsilon$ on $\mathcal{X}$. Then using Proposition 11, let $q$ be a polynomial such that $p(V) = q([V_{i_1,j} \cdot V_{i_2,j}]_{i_1 \in [n], i_2 \in [n], j \in [k]})$.

Define $\phi : \mathbb{R}^n \to \mathbb{R}^{n^2}$ to map a $v \in \mathbb{R}^n$ to the vector of pairwise products of elements in $v$ scaled by $1/2$, that is

$$\phi(v) = \frac{1}{2} \text{vec}(vv^\top) \tag{93}$$

Then $\phi(v) + \phi(-v)$ is equal to the vector of pairwise products of $v$. Finally, we let $\rho = q$, which gives that

$$p(V) = \rho([\phi(v_i) + \phi(-v_i)]_{i=1,\ldots,k}), \tag{94}$$

and hence

$$|f(V) - \rho([\phi(v_i) + \phi(-v_i)]_{i=1,\ldots,k})| = |f(V) - p(V)| < \epsilon \tag{95}$$

for all $V \in \mathcal{X}$. $\qquad\square$

Given the form of the sign invariant polynomials, this proof is quite simple. However, it is technically weaker than the result of Lim et al. [2023], as they invoke the Strong Whitney Embedding Theorem and only require $\phi$ to map to $\mathbb{R}^{2n}$ instead of $\mathbb{R}^{n^2}$. Still, further arguments could probably reduce the dimension required to about $2n$ in this polynomial-based proof; as the Gram matrix $vv^\top$ is rank one, it can be recovered almost always from about $2n$ of its entries [Pimentel-Alarcón et al., 2016].

# D   Sign Equivariant Architecture Universality

In this section, we prove Proposition 3 on the universality of our proposed sign equivariant architectures, which we restate here:

**Proposition 3.** *Functions of the form $v \mapsto v \odot \mathrm{MLP}(|v|)$ universally approximate continuous sign equivariant functions $f : \mathbb{R}^k \to \mathbb{R}^k$.*

*Compositions $f_2 \circ f_1$ of functions $f_l$ as in equation 6 universally approximate continuous sign equivariant functions $f : \mathbb{R}^{n \times k} \to \mathbb{R}^{n' \times k}$.*

We prove the two statements of the proposition in the next two subsections.

## D.1 Universality for functions $\mathbb{R}^k \to \mathbb{R}^k$

*Proof.* Let $\mathcal{X} \subseteq \mathbb{R}^k$ be a compact set, let $\epsilon > 0$, and let $f_{\mathrm{target}} : \mathcal{X} \to \mathbb{R}^k$ be a continuous sign equivariant function that we wish to approximate within $\epsilon$. Choose a sign equivariant polynomial $p$ that approximates $f_{\mathrm{target}}$ to within $\epsilon/2$ on $\mathcal{X}$. By compactness, we can choose a finite bound $B > 0$ such that $|v_i| < B$ for all $v \in \mathcal{X}$.

By Proposition 9, we can write $p(v)_l = v_l \cdot \sum_{d_1,\ldots,d_k=0}^{D} \mathbf{W}_{d_1,\ldots,d_k} v_1^{2d_1} \cdots v_k^{2d_k}$. By the universal approximation theorem for multilayer perceptrons, we can choose a MLP : $\mathcal{X} \to \mathbb{R}^k$ such that approximates $q(v) = \sum_{d_1,\ldots,d_k=0}^{D} \mathbf{W}_{d_1,\ldots,d_k} v_1^{2d_1} \cdots v_k^{2d_k}$ up to $\epsilon/(2B)$. Note that $q(|v|) = q(v)$, so $v \mapsto \mathrm{MLP}(|v|)$ also approximates $q$ within $\epsilon/(2B)$ accuracy.

Thus, for all $v \in \mathcal{X}$, we have that

$$|f(v)_i - p(v)_i| = |v_i \cdot \mathrm{MLP}(|v|)_i - v_i \cdot \sum_{d=1}^{D} \mathbf{W}_{d_1,\ldots,d_k} v_1^{2d_1} \cdots v_k^{2d_k}| \tag{96}$$

$$= |v_i| |\mathrm{MLP}(|v|)_i - \sum_{d=1}^{D} \mathbf{W}_{d_1,\ldots,d_k} v_1^{2d_1} \cdots v_k^{2d_k}| \tag{97}$$

$$\leq B \cdot |\mathrm{MLP}(|v|)_i - \sum_{d=1}^{D} \mathbf{W}_{d_1,\ldots,d_k} v_1^{2d_1} \cdots v_k^{2d_k}| \tag{98}$$

$$< \epsilon/2, \tag{99}$$

so $\|f - p\|_\infty < \epsilon/2$ on $\mathcal{X}$ and we are done by the triangle inequality. $\square$

## D.2 Universality for functions $\mathbb{R}^{n \times k} \to \mathbb{R}^{n' \times k}$

Recall that each layer of our sign equivariant network from $\mathbb{R}^{n \times k} \to \mathbb{R}^{n' \times k}$ takes the form

$$f_l(V) = [W_1^{(l)} v_1, \ldots, W_k^{(l)} v_k] \odot \mathrm{SignNet}_l(V).$$

*Proof.* Let $\mathcal{X} \subseteq \mathbb{R}^{n \times k}$ be compact, and let $f_{\mathrm{target}} : \mathcal{X} \to \mathbb{R}^{n' \times k}$ be a continuous sign equivariant function that we wish to approximate. Since $\mathcal{X}$ is compact, we can choose a finite bound $B > 0$ such that $|V_{ij}| < B$ for all $V \in \mathcal{X}$. Let $p : \mathcal{X} \subseteq \mathbb{R}^{n \times k} \to \mathbb{R}^{n' \times k}$ be a sign equivariant polynomial that approximates $f_{\mathrm{target}}$ up to $\epsilon/2$ accuracy. Using Proposition 10, we can write

$$p(V)_{a,b} = \sum_{d_{i,j}=0}^{D} \mathbf{W}_{\mathbf{d}}^{(a,b)} V_{i_{\mathbf{d}},b} \cdot p_{\mathrm{inv}}(V)_{\mathbf{d}},$$

for some sign invariant polynomials $p_{\mathrm{inv}}(V)_{\mathbf{d}}$. We will have one network layer $f_1$ approximate the summands, and have the second network layer $f_2$ compute the sum.

First, we absorb the coefficients $\mathbf{W}_{\mathbf{d}}^{(a,b)}$ into the sign invariant part, by defining the sign invariant polynomial $q_{\mathrm{inv}}(V)_{\mathbf{d},a,b} = \mathbf{W}_{\mathbf{d}}^{(a,b)} p_{\mathrm{inv}}(V)_{\mathbf{d}}$, so we can write

$$p(V)_{a,b} = \sum_{d_{i,j}=0}^{D} V_{i_{\mathbf{d}},b} \cdot q_{\mathrm{inv}}(V)_{\mathbf{d},a,b}.$$

Now, let $d_{\mathrm{hidden}} \in \mathbb{N}$ denote the number of all possible $\mathbf{d}$ that appear in the sum, multiplied by $n'$. We define $f_1 : \mathcal{X} \to \mathbb{R}^{d_{\mathrm{hidden}} \times k}$ as follows. As SignNet [Lim et al., 2023] universally approximates

sign invariant functions on compact sets, we can let $\text{SignNet}_1 : \mathcal{X} \to \mathbb{R}^{d_{\text{hidden}} \times k}$ be a SignNet that approximates $q_{\text{inv}}(V)$ up to $\epsilon/(2B)$ accuracy, so

$$|\text{SignNet}_1(V)_{(\mathbf{d},a),b} - q_{\text{inv}}(V)_{\mathbf{d},a,b}| < \frac{\epsilon}{2B \cdot d_{\text{hidden}}}. \tag{100}$$

For $b \in [k]$, we also define the weight matrices $W_b^{(1)} \in \mathbb{R}^{d_{\text{hidden}} \times n}$ of the layer by letting the $(\mathbf{d}, a)$th row $(W_b^{(1)})_{(\mathbf{d},a),:}$ for any $a \in [n]$ only be nonzero in the $i_\mathbf{d}$th index, where it is equal to 1. Thus,

$$(W_b^{(1)} v_b)_{(\mathbf{d},a)} = V_{i_\mathbf{d},b}. \tag{101}$$

Hence, the first layer takes the form

$$f_1(V)_{(\mathbf{d},a),:} = \begin{bmatrix} V_{i_\mathbf{d},1} \cdot \text{SignNet}_1(V)_{(\mathbf{d},a),1} & \cdots & V_{i_\mathbf{d},k} \cdot \text{SignNet}_1(V)_{(\mathbf{d},a),k} \end{bmatrix} \in \mathbb{R}^k. \tag{102}$$

Now, for the second layer, we let $\text{SignNet}(V)_{i,j} = 1$ for all $i \in [n], j \in [k]$, which can be represented exactly. Then for each column $b \in [k]$ we will define weight matrices $W_b^{(2)}$ such that $(W_b^{(2)})_{a,(\mathbf{d},i)} = 1$ if $a = i$ and is 0 otherwise. Then we can see that

$$f_2 \circ f_1(V)_{a,b} = \sum_\mathbf{d} V_{i_\mathbf{d},b} \cdot \text{SignNet}_1(V)_{(d,a),b}. \tag{103}$$

To see that this approximates the polynomial $p$, for any $V \in \mathcal{X}$ we can bound

$$|p(V)_{a,b} - f_2 \circ f_1(V)_{a,b}| = \left| \sum_\mathbf{d} V_{i_\mathbf{d},b} \cdot \left( q_{\text{inv}}(V)_{\mathbf{d},a,b} - \text{SignNet}_1(V)_{(\mathbf{d},a),b} \right) \right| \tag{104}$$

$$\leq \sum_\mathbf{d} |V_{i_\mathbf{d},b}| \left| \left( q_{\text{inv}}(V)_{\mathbf{d},a,b} - \text{SignNet}_1(V)_{(\mathbf{d},a),b} \right) \right| \tag{105}$$

$$\leq B \sum_\mathbf{d} \left| \left( q_{\text{inv}}(V)_{\mathbf{d},a,b} - \text{SignNet}_1(V)_{(\mathbf{d},a),b} \right) \right| \tag{106}$$

$$< B \sum_\mathbf{d} \frac{\epsilon}{2B d_{\text{hidden}}} \tag{107}$$

$$\leq \frac{\epsilon}{2} \tag{108}$$

By the triangle inequality, $f_2 \circ f_1$ is $\epsilon$-close to $f_{\text{target}}$, so we are done.

$\square$

# E  Experimental Details

## E.1  Miscellaneous Experimental Details

We ran the experiments on a HPC server with CPUs and GPUs. Each experiment was run on a single NVIDIA V100 GPU with 32GB memory. The runtimes for some of our experiments are included in the main paper. Our codes for our models and experiments will be open-sourced and permissively licensed.

## E.2  Link Prediction in Nearly Synthetic Graphs

The base graphs $H$ we generate are Erdös-Renyi or Barabási-Albert graphs with 1000 nodes. We use NetworkX [Hagberg et al., 2008] to generate and process the graphs. The Erdös-Renyi graphs have edge probability $p = .05$ and the Barabási-Albert graphs have $m = 20$ new edges per new node. Let $V = [v_1, \ldots, v_k]$ be Laplacian eigenvectors of the graph. We take $k = 16$ in these experiments. The unlearned decoder baseline simply takes the predicted probability of a link between $i$ and $j$ to be proportional to the dot product of the eigenvectors embeddings of node $i$ and node $j$; this has no learnable parameters. In other words, the node embeddings $z_i$ and $z_j$ are taken to be $V_{i,:}$ and $V_{j,:}$ respectively, and the edge prediction is $z_i^\top z_j$. The learned decoder baseline takes the same $z_i$ and $z_j$,

but takes the edge prediction to be $\mathrm{MLP}(z_i \odot z_j)$. Every other method learns node embeddings $z_i$ and $z_j$, and takes the edge prediction to be $z_i^\top z_j$.

Each model is restricted to around 25,000 learnable parameters (besides the Unlearned Decoder, which has no parameters). We train each method for 100 epochs with an Adam optimizer [Kingma and Ba, 2015] at a learning rate of .01. The train/validation/test split is 80%/10%/10%, and is chosen uniformly at random.

### E.3 Details on n-body Simulations

We follow the experimental setting and build on the code of Puny et al. [2022] (no license as far as we can tell) for the n-body learning task. The code for generating the data stems from Kipf et al. [2018] (MIT License) and Fuchs et al. [2020] (MIT License). There are 3000 training trajectories, 2000 validation trajectories, and 2000 test trajectories. We modify the data generation code to apply to general dimensions $d > 3$. We do not change any of the scaling factors in doing so. For each dimension $d$, we use the same hyperparameters for both the frame averaging model and the sign equivariant model.

Table 5: n-body simulation results for dimension $d = 3$. Lower MSE is better. Results are from [Satorras et al., 2021, Puny et al., 2022, Kaba et al., 2023].

| Model | Test MSE |
|---|---|
| Linear | .0819 |
| $SE(3)$ Transformer [Fuchs et al., 2020] | .0244 |
| TFN [Thomas et al., 2018] | .0155 |
| Radial Field [Köhler et al., 2020] | .0104 |
| EGNN [Satorras et al., 2021] | .0071 |
| FA-GNN [Puny et al., 2022] | .0057 |
| CN-GNN [Kaba et al., 2023] | **.0043** |
| Sign Equivariant (Ours) | .0065 |

In Table

### E.4 Node Classification on CLUSTER

In Section 4.3, we show results for the node classification task CLUSTER [Dwivedi et al., 2022a], where the task is to cluster nodes in graphs drawn from Stochastic Block Models [Abbe, 2017]. Models are restricted to a 100k learnable parameter budget. We largely follow the experimental setting of Rampasek et al. [2022], except we report results for the eigenvector based methods on 5 runs instead of 10.

We test several eigenvector based methods within the GraphGPS framework and codebase [Rampasek et al., 2022] (MIT License), which is a state of the art Transformer / GNN hybrid. Firstly, we make use of the PEG style GraphGPS, which means that the MPNN in the $l$th GraphGPS layer takes as edge features $e_{ij}^{(l)} = \left\| V_i^{(l)} - V_j^{(l)} \right\|^2$, where $V_i^{(l)} \in \mathbb{R}^k$ is the eigenvector embedding of node $i$ in layer $l$. This is fully $O(k)$ invariant (which is much stricter than sign / basis invariance), so we relax this to just be sign invariant in our model by learning a diagonal matrix $D^{(l)}$ such that $e_{ij}^{(l)} = V_i^{(l)\top} D^{(l)} V_j^{(l)}$. Also, the standard GraphGPS only updates eigenvector representations (in a non-equivariant manner) before most of the neural network modules. When we add our sign equivariant model, we instead update eigenvector representations within each GraphGPS layer via $V^{(l)} = f_\theta^{(l)}(V^{(l-1)})$ for a sign equivariant $f_\theta^{(l)} : \mathbb{R}^{n \times k} \to \mathbb{R}^{n \times k}$.

