# OpenReview forum: "Expressive Sign Equivariant Networks for Spectral Geometric Learning"
_NeurIPS.cc/2023/Conference — NeurIPS 2023 spotlight_

### Official Review · Reviewer_nERC · 2023-06-16

**Soundness:** 3 good
**Presentation:** 4 excellent
**Contribution:** 3 good
**Rating:** 7
**Confidence:** 3

**Summary:**

The authors proposed a novel sign-equivariant model that is equivariant to the sign flip of eigenvectors for spectral graph learning. They demonstrated that the proposed architecture could capture the information when sign-invariant models would fail. The authors then made theoretical analyses of the sign-equivariant polynomial space and also proposed some constructible neural network architecture with sign-equivariance guaranteed by design. Experimental results on both the synthetic and real-world datasets demonstrate superior performance on link prediction, especially when automorphism exists.

**Strengths:**

1. The paper is well-organized and easy to follow. The authors first provided an example to demonstrate the limitation of the previous "sign-invariant" models and then made theoretical analyses. Experiments were followed to demonstrate superior performance over the baselines.

2. The problem that the authors tried to address is novel. The authors pointed out the limitation of sign-invariant models and mathematically supported their claim in Proposition 1. The proposed "sign-equivariant" architecture can properly address the limitation, both theoretically and empirically.

3. The mathematics are solid and the claims are well-supported. The definitions of sign-equivariance and -invariance are well-formulated. Sign-equivariance of polynomials and neural networks is proven in the appendices. Universality is also proven and guaranteed.

4. Experiments on graphs with high symmetries demonstrated the effectiveness of the proposed model architecture. State-of-the-art performance was also achieved.

**Weaknesses:**

1. Proposition 1 lacks some detail (See Question 1 & 2).
2. Limitation of the proposed architecture is not explicitly discussed.

**Questions:**

1. In Proposition 1, the first result claims that if $f$ is sign invariant and the eigenvalues are distinct, then the node-wise representations are the same for automorphic nodes. Consider the complete graph of 3 nodes, the adjacency matrix is
$$
\begin{pmatrix}
0&1&1\\\\
1&0&1\\\\
1&1&0\\\\
\end{pmatrix}
$$
We have eigenvector $v_1=(1,1,1)$ belonging to $\lambda=2$ and $v_2=(0,1,-1)$ belong to $\lambda=-1$ (normalization is ignored for conciseness). Let $f$ be the pointwise absolute function, then $f$ is clearly sign-invariant (but not necessarily basis-invariant). In this graph, all three nodes are automorphic, but we have $z_1=(1,0)$ and $z_2=(1,1)$. This is probably because the eigenspace corresponding to $\lambda=-1$ is 2-dimensional. Therefore, I believe basis-invariance is also a necessary condition. The first claim is also not directly proven in Appendix A.3.2. Please provide a more detailed justification.

2. Similarly, the scenario when the dimension of the eigenvalue is greater than 1 is not explicitly discussed. In the above example, the sign-invariant model may still be capable of distinguishing the automorphic nodes. In footnote 2, the authors mentioned that sign invariant embeddings maintain "some positional information". The authors may provide a more accurate description using mathematical formulations similar to Proposition 1.

3. What are the potential limitations of the proposed model architecture?



**Limitations:**

No potential negative societal impact is expected.

---

> ### Author Rebuttal · Authors · 2023-08-08
>
> > “Proposition 1 lacks some detail (See Question 1 & 2).”
>
> > “In Proposition 1, the first result claims that if $f$ is sign invariant and the eigenvalues are distinct, then the node-wise representations are the same for automorphic nodes. Consider the complete graph of 3 nodes, the adjacency matrix is…”
>
> Indeed, this issue that you note here arises because the eigenspace corresponding to $\lambda=-1$ is two-dimensional. This misunderstanding is because we do not exactly define what we mean in Proposition 1 when we say “the eigenvalues associated with the $v_l$ are distinct.” By this, we mean that the eigenvalues $\lambda_l$ are all simple eigenvalues, meaning that they all belong to one-dimensional eigenspaces, and also no two of the eigenvectors belong to the same eigenvalue. We will make this clear in the revision.
>
> With this correct interpretation of our assumptions, we are confident that the proposition is correct in the sign invariance case: we only need that the eigenvalues corresponding to the input $v_1, \ldots, v_k$ are all simple eigenvalues. The proof of the sign invariant case is exactly the same as the general basis invariant case; the orthogonal matrices $Q_t$ are just $1$ dimensional and hence they are signs.
>
> > “Similarly, the scenario when the dimension of the eigenvalue is greater than 1 is not explicitly discussed. In the above example, the sign-invariant model may still be capable of distinguishing the automorphic nodes. In footnote 2, the authors mentioned that sign invariant embeddings maintain "some positional information". The authors may provide a more accurate description using mathematical formulations similar to Proposition 1.”
>
> True, the repeated eigenvalues do arise in practice, and sign invariant embeddings are partially positional in this case since they are not fully basis invariant. It is hard to say too much in this setting, since the sign invariant embeddings are not even well-defined / deterministic here (the embeddings would depend on the choice of basis of the eigenspaces used as input). This also means that sign invariant embeddings would not lead to structural node-pair representations when there are repeated eigenvalues. We will make this latter point in the revision.
>
> > “Limitation of the proposed architecture is not explicitly discussed.”
> > “What are the potential limitations of the proposed model architecture?”
>
> Thanks for the suggestion, see general comment to all reviewers for discussion of limitations.

---

> > ### Comment · Reviewer_nERC · 2023-08-18
> > **Comment on Authors' Rebuttal**
> >
> > I appreciate your comprehensive response regarding my concerns about some theorems on sign equivariance. You have also frankly and meticulously explained the potential limitation of the proposed model when the eigenvalues coincide. Though the proposed model may fail in this case, to the best of my knowledge, I believe that the authors have paved the road for a new sub-realm for further work on sign equivariance, which is important for tasks like link prediction where node-level GNNs would fail. In this sense, **I raise my score from 6 to 7**.

---

### Official Review · Reviewer_p512 · 2023-07-06

**Soundness:** 4 excellent
**Presentation:** 4 excellent
**Contribution:** 3 good
**Rating:** 8
**Confidence:** 3

**Summary:**

The paper proposes a sign-equivariant design for processing spectral features in geometric deep learning.
This is particularly useful to process the eigenvectors of graph Laplacians and generate graph positional encodings, especially for link prediction tasks.
The equivariant, rather than invariant, approach guarantees improved expressivity of the models, which is verified in a numberof experiments.


**Strengths:**

The paper is clearly written and well motivated.
I liked this simple yet effective design and I think the proposed method could be useful for many future works in the field.

**Weaknesses:**

I don't see major flaws in the paper.
See "Questions" for other comments.


**Questions:**

I don't think the claim that the Geometric Deep Learning approach fails here is correct.
The issue you describe is due to a bad design choice of intermediate representation within the network (i.e. the same as the input and output representation).
Note that this choice is not common in other equivariant networks either; for instance, in group-convolution NNs (GCNNs), the input representation is an image (this is a quotient representation of $SE(2)$) but intermediate features transform like a regular representation (the output of a group convolution is a function over the full group, not only the pixels).
This step is fundamental to achieve universal approximation.
The limited expressive power you mention in line 48 is only due to the choice of intermediate representations (similarly, in a GCNN, if the intermediate features are constrained to transform as images, equivariance requires isotropic filters, which are indeed less expressive).

In any case, while a group-convolution approach is probably a poor choice here (the group $(-1, 1)^k$ has size $2^k$), smaller representations which can provide a better trade off might exist.
Your method seems to provide a nice solution to this, so I would argue it still fits within the GDL framework but provides a more efficient solution than its naive implementation.


Isn't computing the eigenvectors of the graph Laplacian an expensive preprocessing step?
Estimating these features has $O(n^3)$ complexity, which seems to exceed the $O(n)$ operations in the DSS layers (or $O(n^2)$ is an attention-based or message-passing based model).
In Appendix A.2 you discuss the computational complexity of your method but do not include this aspect.
Could you comment on this?

Could you report the table with all baselines and their performance in Section 4.2?
Also, is there any more recent work on this dataset you could include?


**Limitations:**

I did not see an explicit discussion of the limitations in the paper, which I think would be a nice addition.
For example, the authors could comment further on the complexity of computing the eigenvectors.

---

> ### Author Rebuttal · Authors · 2023-08-08
>
> We thank the reviewer for their thoughtful comments on the choice of group representations in intermediate layers. Here, we respond to the comments; we will add discussion about this to the main paper, which we think will be interesting to people in the equivariant machine learning community.
>
> > “I don't think the claim that the Geometric Deep Learning approach fails here is correct. The issue you describe is due to a bad design choice of intermediate representation within the network … “
>
> The reviewer makes a great point here, the claim is more nuanced than how we wrote it in this paper. We will change our wording of the point to be less strong and capture more of the nuance (namely, the approach does not work for some choices of group representation, but other group representations may allow it to succeed). We also had some more results that we did not end up adding to the paper, which we now plan to add to support this point a bit more.
>
> In particular, we derived the sign equivariant linear maps between tensor representations, and found that it would not be efficient to use tensor representations either. That is, at $n=1$ we considered the sign equivariant maps from $\mathbb{R}^{k^{d_1}} \to \mathbb{R}^{k^{d_2}}$ for $d_1, d_2 > 0$. The sign group acts on $\mathbb{R}^{k^{d_1}}$ via $((s_1, \ldots, s_k) \cdot x)_{i\_1, \ldots, i\_{d_1}} =s\_{i\_1} \cdots s\_{i\_{d_1}} x\_{i\_1, \ldots, i\_{d\_1}}$. Note that the $d_1 = d_2 = 1$ case is the one considered in the current paper, where the only equivariant linear maps are diagonal matrices, so the equivariant linear maps have dimension $k$.
>
> We can show that there are no sign equivariant linear maps (besides the zero map) between $\mathbb{R}^{k^{d_1}} \to \mathbb{R}^{k^{d_2}}$ when $d_1 + d_2$ is an odd number. Note that this is also the case for the orthogonal group, as can be seen numerically in Table 6 of Finzi et al. 2021. In particular, this means that there is no linear way to map $\mathbb{R}^{k} \to \mathbb{R}^{k^2}$ or back $\mathbb{R}^{k^2} \to \mathbb{R}^k$, so if we want to lift our $\mathbb{R}^{k}$ input to a tensor representation, you have to at least go to order 3 tensors (in $\mathbb{R}^{k^3}$). This would already be quite expensive, and the linear maps between order 3 tensors ($\mathbb{R}^{k^3} \to \mathbb{R}^{k^3}$) are also expensive, since they span a space of dimension 5888 when $k=10$.
>
> This made us feel that the equivariant linear maps approach with tensor representations is too inefficient, so we did not pursue it further. We will add proofs of these points in the revised version, as others may find it interesting or useful.
>
> > “smaller representations which can provide a better trade off might exist. Your method seems to provide a nice solution to this, so I would argue it still fits within the GDL framework but provides a more efficient solution than its naive implementation.”
>
> This is a good point! As argued above, the natural tensor representations (as used e.g. in [Maron et al. 2019] https://arxiv.org/abs/1812.09902 and [Finzi et al. 2021] https://arxiv.org/abs/2104.09459) are not a good choice for equivariant linear map based architectures for the sign group. We agree with you that regular representations of size $2^k$ are also too expensive without any tricks. However, other representations may be useful, and we will note this possibility in the main paper.
>
> > “Isn't computing the eigenvectors of the graph Laplacian an expensive preprocessing step? … Estimating these features has $O(n^3)$ complexity…”
>
> As is commonly done in practice, we only use some subset of the Laplacian eigenvectors corresponding to the smallest $k$ eigenvalues (e.g. $k=16$ for our link prediction experiments). Then standard iterative eigensolvers can be used (e.g. scipy.sparse.linalg.eigsh), which are very efficient for sparse graphs. The time complexity is closer to $O(|E| k)$, where $|E|$ is the number of edges, which is usually close to linear in the number of nodes. For instance, on a standard laptop, computing the smallest $k=16$ eigenvectors of an Erdos-Renyi graph with average degree 10 takes 0.3 seconds for 10,000 nodes and 9.0 seconds for 100,000 nodes.
>
> For the n-body experiments, the covariance matrix is $d \times d$, where $d$ is the ambient dimension, which is typically quite small. Thus, there are no efficiency issues here as well.
>
> > “Could you report the table with all baselines and their performance in Section 4.2? Also, is there any more recent work on this dataset you could include?”
>
> As suggested by Reviewer YXFy, we can also add the Kaba et al. paper just published at ICML 2023 https://arxiv.org/abs/2211.06489; this method achieves .0043 MSE in $d=3$ dimensions (hence outperforming our method in accuracy here), but the runtime scaling in $d$ may be worse, since their canonicalization network has to output a $d \times d$ matrix that must then be orthogonalized via Gram-Schmidt, which has $O(d^3)$ complexity. Further, they do not test their methods on $d > 3$, and we do not have access to their code.
>
> For the other baselines, we will include a table (possibly in the Appendix due to space constraints), but this would only include $d=3$ (as many methods only test on $d=3$, or only work on $d=3$).
>
> > “I did not see an explicit discussion of the limitations in the paper, which I think would be a nice addition. For example, the authors could comment further on the complexity of computing the eigenvectors.”
>
> Thank you for the suggestion, see the general comment to all reviewers for limitations.

---

> > ### Comment · Reviewer_p512 · 2023-08-15
> >
> > I thank the authors for the detailed answer.
> >
> > I have also appreciated the discussion regarding my first question about the GDL framework and encourage the authors to include it in the paper.
> > In particular, I am very interested in reading the theoretical results about linear equivariant maps between tensor product representations.
> >
> > I maintain my recommendation

---

### Official Review · Reviewer_a3kf · 2023-07-07

**Soundness:** 4 excellent
**Presentation:** 4 excellent
**Contribution:** 3 good
**Rating:** 7
**Confidence:** 3

**Summary:**

This paper contributes construction and analysis of sign equivariant neural network architectures for processing eigenvectors while respecting their symmetries. While a similar approach has been proposed by a prior work (Lim et al., 2023), this work is motivated by the fact that sign invariance, pursued in prior work, is in general not sufficient for multi-node tasks as sign invariance leads to structural node representations that does not distinguish automorphic nodes while multi-node tasks require positional node representations that provides a form of node identification (Srinivasan et al., 2019). The authors provide a construction of sign equivariant neural network architecture with analysis and provable guarantee on expressive power, and also propose its application as an alternative to PCA-based frame averaging which requires only a single evaluation of base model (while frame averaging requires 2^k) while guaranteeing equivariance to orthogonal groups in arbitrary dimensions. The developed architectures are empirically demonstrated for link prediction on graphs, n-body dynamics prediction, and node clustering on SBM graphs, which support the main claims of the paper.

Lim et al., Sign and Basis Invariant Networks for Spectral Graph Representation Learning (2023)

Srinivasan et al., On the equivalence between positional node embeddings and structural graph representations (2019)

**Strengths:**

S1. The paper tackles an important and practical problem of developing expressive neural architectures that respect symmetry of eigenvectors, and successfully complements and improves key prior work (Lim et al., 2023) by achieving sign equivariance. Also, the paper is overall well written and easy to read, and the provided illustrations are clear.

S2. The motivation for sign equivariance over invariance only is clearly presented in Section 2.1 where multi-node tasks are considered, and I find the discussion sound and informative.

S3. In addition to multi-node tasks, a novel (as far as I can tell) and powerful alternative based on sign equivariance to PCA based frame averaging for modeling O(k) equivariant functions is provided. The approach trades off generality, as the base model (h) should satisfy sign equivariance unlike frame averaging that works with arbitrary base functions, but more efficient as it eliminates the requirement of frame averaging for 2^k forward passes.

S4. The proposed parameterization of sign equivariant polynomial is novel and theoretically sound as it is grounded to the theory of equivariant polynomials. A strong theoretical guarantee on universality is provided for the non permutation equivariant case as well.

S5. Overall, the experimental results overall seem sound and support the main claims of the paper.

Lim et al., Sign and Basis Invariant Networks for Spectral Graph Representation Learning (2023)

**Weaknesses:**

W1. The description of how sign equivariant network is applied in Appendix D.4 was a bit unclear to me. In in the equation e_ij = V_i^TDV_j, does the V refer to the output of sign equivariant module within the given GraphGPS layer? Is V passed and updated across GraphGPS layers?

**Questions:**

I have no specific questions for now other than W1.

**Limitations:**

As far as I can tell, the authors didn't address the limitations and potential societal impact of the work. I encourage the authors to discuss them in the next revision of the paper.

---

> ### Author Rebuttal · Authors · 2023-08-08
>
> We thank the reviewer for their comments. We are especially glad that the reviewer liked section 2.1 on multi-node representations and link prediction, as we originally had trouble writing it, but spent time iterating on it.
>
> > “W1. The description of how sign equivariant network is applied in Appendix D.4 was a bit unclear to me. In in the equation e_ij = V_i^TDV_j, does the V refer to the output of sign equivariant module within the given GraphGPS layer? Is V passed and updated across GraphGPS layers?”
>
> Thank you for pointing this out. Yes, $V \in \mathbb{R}^{n \times k}$ are eigenvector representations, which we compute using our sign equivariant module. This $V$ is passed and updated in each GraphGPS layer using our module.
>
> To make this clearer, in the revision we will denote the eigenvector representation of layer $l$ as $V^{(l)}$. Then $V^{(0)}$ are the original eigenvectors, and $V^{(l)} = \mathrm{SignEquivariant}^{(l)}(V^{(l-1)})$. In contrast, the prior work PEG [Wang et al. 2022] takes $V^{(l)} = V^{(l-1)}$, meaning they do not update eigenvector representations.
>
> > “As far as I can tell, the authors didn't address the limitations and potential societal impact of the work. I encourage the authors to discuss them in the next revision of the paper.”
>
> Thanks for the suggestion, we include a discussion in the general reviewer comment, and will add discussion to our paper.

---

> > ### Comment · Reviewer_a3kf · 2023-08-18
> > **Response to rebuttal**
> >
> > Thank you for the response. I have no questions for now and will discuss with other reviewers.

---

### Official Review · Reviewer_HW6g · 2023-07-08

**Soundness:** 3 good
**Presentation:** 3 good
**Contribution:** 3 good
**Rating:** 6
**Confidence:** 4

**Summary:**

This paper focuses on addressing the sign ambiguity problem of eigenvectors. It argues that previous sign-invariant models are insufficient for some applications, e.g., link prediction and multi-node tasks. To solve this problem, the authors propose a sign-equivariant neural network with provable expressiveness guarantees, which is more powerful than the sign-invariant models and sign-equivariant linear maps. Experiments on various tasks validate the superiority of the proposed method.

**Strengths:**

1. This paper is the first to study the sign-equivariance problem of eigenvectors. It finds that the signs of eigenvectors contain meaningful positional information but the sign-invariant models can only preserve the structural information. Therefore, the sign-invariant models cannot distinguish automorphic nodes and are insufficient for link prediction or multi-node tasks.

2. This paper proves that the sign equivariant polynomial can be implemented by the elementwise product between a linear sign-equivariant polynomial and a general sign-invariant polynomial. The linear sign-equivariant polynomial cannot learn the interactions between eigenvectors, and the sign-invariant polynomial cannot preserve positional information. This theoretical discovery combines the advantages of both methods and addresses their shortcoming.

3. Based on the characterization of sign-equivariant polynomial functions, this paper provides a general framework for analyzing the sign-symmetry models and gives a new perspective on the universality of SignNet.

**Weaknesses:**

The major weakness of this paper is that it lacks experiments on real-world data. This paper uses three tasks, e.g., link prediction, n-body problem, and node clustering, to validate the effectiveness of the proposed method. However, all the experiments are based on synthetic data, which makes the results unconvincing. I think there are many real-world datasets for both link prediction and node clustering tasks, and the authors should give more empirical results on at least these two tasks.

**Questions:**

This paper considers both sign-equivariance and permutation-equivariance. I wonder why the authors do not discuss the basis-equivariance for eigenvectors.

**Limitations:**

This paper should discuss the limitation of sign-equivariance from the perspectives of efficiency and scalability.

---

> ### Author Rebuttal · Authors · 2023-08-09
>
> We thank the reviewer for their work in improving our paper! We are glad that the reviewer appreciates our theoretical characterization of sign equivariant functions and improvements over sign invariant networks. Here we address the comments:
>
> > “The major weakness of this paper is that it lacks experiments on real-world data. This paper uses three tasks, e.g., link prediction, n-body problem, and node clustering, to validate the effectiveness of the proposed method. However, all the experiments are based on synthetic data …”
>
> True, even though the n-body task and node clustering tasks are standard benchmarks, they do consist of synthetic data. We did not manage to run more experiments for the rebuttal, but we believe that our experiments do support our theoretical insights, and note that other reviewers appreciate our experiments.
>
> > “This paper considers both sign-equivariance and permutation-equivariance. I wonder why the authors do not discuss the basis-equivariance for eigenvectors.”
>
> This is a good point and a good idea for future work. The neural architecture would be more difficult to derive for the continuous symmetries in basis equivariance, and it would probably require different methods than our sign equivariant networks, which is why we do not cover it much. Basis equivariance would indeed be useful to handle repeated eigenvalues. For instance, basis equivariance could be used in Section 2.1 to obtain structural node-pair representations when the graphs have repeated eigenvalues. We will add discussion of this latter point to the revised paper.
>
> > “This paper should discuss the limitation of sign-equivariance from the perspectives of efficiency and scalability.”
>
> We are not sure what the reviewer means here, please let us know if we have misunderstood. In our paper, we explain that for both of our main application areas, sign equivariant networks have efficiency and scalability benefits over certain baselines. In link prediction, sign equivariant networks only require one forward pass whereas most subgraph-based methods need a forward pass and graph construction for each predicted edge (Section 2.1, last paragraph). For orthogonal equivariance, sign equivariant networks only require one forward pass for inference on $d$-dimensional point clouds, whereas frame averaging requires $2^d$ forward passes (see also Figure 4 and Section 2.2, paragraph before the Proposition). See also Appendix A.2, A.2.1, and A.2.2 for discussion of complexity.
>
> As for general limitations, we will add them to the revision, see general reviewer comment for more information.

---

> > ### Comment · Reviewer_HW6g · 2023-08-15
> > **Response to rebuttal**
> >
> > Hi, thanks for addressing my concerns.
> >
> > After carefully checking the appendix, I think the sign-equivariance networks have some benefits in terms of both efficiency and scalability.
> >
> > Also, I agree with the authors that the sign-equivariant layers can improve the sign-invariant readout functions, e.g., SignNet. The combination of local equivariant and global invariant is important for geometric deep learning.
> >
> > Although it is a pity that the authors do not provide more empirical results, I think the clear motivation and important theoretical results deserve a good point. Therefore, I raise my score to 6.

---

### Official Review · Reviewer_YXFy · 2023-07-27

**Soundness:** 3 good
**Presentation:** 3 good
**Contribution:** 3 good
**Rating:** 6
**Confidence:** 4

**Summary:**

This paper proposes to build sign equivariant neural networks with applications of the method in O(n) equivariant modeling and graph representation learning. They show that corresponding equivariant MLPs are inexpressive, by contrast to the proposed method which is universal (non-permutation equivariant version) or plausibly much more expressive (permutation equivariant version). Experiments are performed on artifcial datasets and show that the proposed method achieve comparable or better results than alternatives.

**Strengths:**

This is a well written and motivated paper. I also appreciate the detailed appendix.

The experimental protocol seems rigorous and results are presented for a few different settings.

The paper highlights interesting and potentially useful applications of sign (O(1)) equivariance.

**Weaknesses:**

(W1) I think the main result of this paper could be obtained from the results of [Villar 2021]. They already show how to use invariants to builds equivariant functions. Sign equivariance is O(1) equivariance, so it is a particular case of O(n) equivariance. This connection is not mentioned by the authors and would be important to investigate. As such polynomials may not really be needed, the results from invariant theory could be sufficient. Also, Proposition 11 from the aforementioned work may offer a way to obtain the desired permutation equivariance with universality.

(W2) More generally, given the previous comment, I feel like the paper should situate itself more within the literature on O(n) equvariance, since sign and basis equivariance are that. I appreciate that some of the applications considered included processing eigenvectors, which is different from most of what is done in that literature, but this was already put forward by [Lim 2023].

(W3) The experiment on CLUSTERS is interesting but the improvement to other methods is not too important and has not convinced me of the practical usefulness of the method. Especially, since CLUSTERS is a synthetic task. It would also be great to have results on real-world graphs to see if the method brings a performance increase there.

(W4) It seems like the proposal for orthogonal equivariance is a realization of the partial canonicalization framework of [Kaba 2023]. Partial canonicalization is performed with respect to SO(3), with the reminder handled by a sign equivariant function. Proposition 2 follows from their results. The results on the N-body problem seem worst compared to learned canonicalization, which should be included for comparison. There are also many other methods outside of those presented in [Puny 2022] that could be compared against.

(W5) The paper [Sachs 1983] is relevant to the Proposition 1 presented in this paper and should be mentioned.

(W6) I think the application on link prediction is the most promising aspect of the submission. The authors should compare with the setup/methods that was proposed by [Sartorras 2021] in their autoencoding experiment. In that paper, the method was used to solve a similar issue caused by automorphic nodes.

**Questions:**

(Q1) Can the authors provide a comparaison of their theoretical results with [Villar 2021] and discuss how their work compares?

(Q2) Would you elaborate on some limitations of this work and potential future directions?

**Limitations:**

Yes

---

> ### Author Rebuttal · Authors · 2023-08-10
>
> We thank the reviewer for their in-depth comments and suggestions for our paper! Here we address the comments:
>
> > “(W1) I think the main result of this paper could be obtained from the results of [Villar 2021]. ... Sign equivariance is O(1) equivariance, so it is a particular case of O(n) equivariance… ”
>
> We kindly disagree. The key difference is that they only consider $O(d)$ (or $O(1)$), whereas the main group we consider is a direct product $O(1) \times \ldots \times O(1)$ (for sign equivariance) or $O(d_1) \times \ldots \times O(d_l)$ (for basis invariance).
>
> These are related, but the direct product group has specific challenges of its own. For instance, interactions between eigenspaces cannot be obtained by simple methods like equivariant linear maps (as explained in Section 3.1).
>
> We are not aware of any way to directly obtain our results using the results of [Villar et al. 2021]. But we are open to more specific suggestions if the reviewer has any.
>
> > ”such polynomials may not really be needed, the results from invariant theory could be sufficient”
>
> We are not aware of a result in invariant theory from which our results follow directly. Invariant theory is (in large part) the study of equivariant or invariant polynomials, and our derivations allow us to find a form specifically for the sign equivariant polynomials.
>
> > “Also, Proposition 11 from the aforementioned work may offer a way to obtain the desired permutation equivariance with universality.”
>
> Thanks for the suggestion. We were very much aware of this work and these results, but we did not figure out a way to use them for our case. Also, this result does not lead to an efficient network architecture (directly parameterizing the functions would require $nd$ neural networks, with permutation constraints that must be enforced as in equation 13 of their arxiv version).
>
> > “(W2) … paper should situate itself more within the literature on O(n) equvariance, since sign and basis equivariance are that. I appreciate that some of the applications considered included processing eigenvectors … but this was already put forward by [Lim 2023].”
>
> Sign and basis equivariance are not quite $O(d)$ equivariance, but rather $O(d_1) \times \ldots \times O(d_k)$ equivariance (as mentioned above). They are related though, and we will include more information in the revision about the connection to $O(d)$ equivariance literature. At the moment we have some relevant content on this in Appendix A.2.1. Some key differences in our case are the proof techniques (the direct product adds complexity, but the fact that we mostly consider $d=1$ for the sign group allows us to use simpler linear algebraic techniques).
>
> We indeed do process eigenvectors as in [Lim 2023], but as noted in Sections 2.1 and 2.2, sign invariance [Lim 2023] is provably limited in learning multi-node representations or $O(d)$ equivariant functions, whereas our sign equivariance methods are provably expressive here.
>
> > “(W4) It seems like the proposal for orthogonal equivariance is a realization of the partial canonicalization framework of [Kaba 2023] ... Proposition 2 follows from their results. The results on the N-body problem seem worst compared to learned canonicalization, which should be included for comparison … .”
>
> Thanks for this, you are correct in that our method works within the [Kaba et al. 2023] framework of partial canonicalization, in which our subgroup is $K = \\{-1, 1\\}^d$ of the orthogonal group $G = O(d)$. Moreover, our Proposition 2 does follow from their Theorems 3.1 and 3.3. We will note this nice connection in our revision, though we will keep our own proof as well for clarity.
>
> Our novel contributions in relation to [Kaba et al. 2023] are:
>
> 1. We choose $K = \\{-1, 1\\}^d$. This is an important design choice, and Kaba et al only choose the trivial group $K = I$ when dealing with $G = O(d)$ in practice.
> 2. We choose an unlearned canonicalization up to $K$, via PCA frames [Puny et al. 2022]. Kaba et al. achieve their best results with learned canonicalizations.
> 3. We are the first to design a $K = \\{-1, 1\\}^d$ equivariant network. This is a necessary component, and we are also able to do this in a provably universal way, which is required to apply Theorem 3.3 of Kaba et al.
>
> We will add discussion of this to the revision, and we will add comparison to more recent $O(d)$ equivariant models like Kaba et al.
>
> > “(W5) The paper [Sachs 1983] is relevant to the Proposition 1 presented in this paper and should be mentioned.”
>
> We assume you are referring to the paper “Automorphism group and spectrum of a graph.“ In our opinion it is not necessarily so relevant to our Proposition 1, in the sense that one can understand either without understanding the other.
>
> However, what is certainly relevant is the general connection between graph automorphisms and repeated eigenvalues. We will add some discussion to our paper about this. In summary, graph automorphisms often lead to repeated eigenvalues, but not always. We will add this reference and discussion to our paper.
>
> > “(W6) I think the application on link prediction is the most promising aspect of the submission. The authors should compare with the setup/methods that was proposed by [Sartorras 2021] in their autoencoding experiment ...”
>
> Great point, perhaps our method could be used for their autoencoding experiment, but we did not have time to try it; our method would take eigenvectors as additional node features to break node symmetry (instead of random noise), and decode in a similar sign invariant way. This is nice, as our method would learn structural node-pair representations if eigenvalues are distinct, whereas their method is not exactly permutation equivariant (since they use random noise). We may try this experiment in a later version of our paper.
>
> > “(Q1) Can the authors provide a comparaison of their theoretical results with [Villar 2021] ...”
>
> Yes, see above. We will add further discussion to our paper.

---

> > ### Comment · Reviewer_YXFy · 2023-08-16
> >
> > I thank the authors for their detailed response. Most of my comments have been addressed, although I still think a bit more results on the experimental side will strengthen the paper significantly. I was indeed mistaken about the group of interest, or at least underestimated the difficulty of tackling the product in a general way. It is worth emphasizing this more explicitly in the paper for clarity. As such, the paper makes a significant contribution. I am therefore happy to increase my score and recommend acceptance.

---

> > > ### Author Response · Authors · 2023-08-16
> > >
> > > Thank you for your reconsideration and reply. Indeed, upon rereading we realize that we can make the symmetry groups under consideration more clear. Thank you for this suggestion, we will do this in the revision.

---

### Author Rebuttal · Authors · 2023-08-08

We would like to sincerely thank all of the reviewers for the work they put into their reviews. The reviews are thoughtful, and the reviewers are each clearly experts in at least some of the several subject areas related to the submission (geometric deep learning, orthogonally equivariant models, spectral graph theory, graph neural networks); we very much appreciate these different viewpoints.

The reviewers suggested adding limitations (and potential societal impacts) several times, so we address them here. We will add these points to the revised version:
1. Limitation: we did not develop architectures for basis equivariance in the case of repeated eigenvalues. Repeated eigenvalues are known to occur in many real-world graphs [Lim et al. 2023], so future work in this area could be useful.
2. Limitation: though we expect the most gain in some node-level and multi-node level tasks on graphs, we do not have theoretical reasons to expect as much impact of sign equivariance on graph-level representations tasks, which for instance is common in molecular processing.
3. Limitation: our theoretical results and model design are focused on expressive power. We do not have results on optimization (e.g. [Xu et al. 2021]), robustness (e.g. [Wang et al. 2022]), or generalization (e.g. [Keriven & Vaiter 2023]). These latter three properties are very important for learning, so future work should address them.
4. Limitation: while we can achieve universal approximation in the non-permutation-equivariant setting, we do not know of the exact expressive power in the permutation-equivariant setting. We can not directly apply existing results from Maron et al. 2020, because the sign group does not act via permutation matrices here. Note that Lim et al. 2023 also faces this issue, and the exact expressive power for their permutation equivariant and sign invariant networks is unknown.
5. Societal impact: we do not foresee direct societal impacts from our work. This project is primarily theoretical and aims to improve models for two general application areas: multi-node representation learning and orthogonal equivariant models. Potential societal impacts may arise in downstream applications that may be affected by general progress in geometric machine learning, such as social network analysis and recommender systems. These two applications are known to have negative societal impacts in certain circumstances, so care must be taken in future related work to avoid major negative consequences.

**References**
[Lim et al. 2023] Sign and Basis Invariant Networks for Spectral Graph Representation Learning.
[Xu et al. 2021] Optimization of Graph Neural Networks: Implicit Acceleration by Skip Connections and More Depth.
[Wang et al. 2022] Equivariant and Stable Positional Encoding for More Powerful Graph Neural Networks
[Keriven & Vaiter 2023] What functions can Graph Neural Networks compute on random graphs? The role of Positional Encoding

---

### Decision · Program_Chairs · 2023-09-21

**Decision:**

Accept (spotlight)

**Comment:**

Reviewers were all positive about the paper, noting that: The problem of sign equivariance is important and well motivated. The paper is well written, well organized and easy to follow. The architecture design is simple yet effective. The mathematics is solid, and provides theoretical (universality) guarantees. The claims of the paper are well supported by experiments, and strong results are obtained. As such I am happy to recommend the paper to be accepted.